# Geographical Differentiation of Mites from the Suborder Uropodina (Acari: Mesostigmata) in Dead Wood in Europe in the Light of Recent Research

Jerzy Błoszyk [1,2], Agnieszka Napierała [1,*] , Marta Kulczak [1] and Michał Zacharyasiewicz [1]

[1] Department of General Zoology, Faculty of Biology, Adam Mickiewicz University in Poznań, Uniwersytetu Poznańskiego 6, 61-614 Poznań, Poland
[2] Natural History Collections, Faculty of Biology, Adam Mickiewicz University in Poznań, Uniwersytetu Poznańskiego 6, 61-614 Poznań, Poland
* Correspondence: agan@amu.edu.pl

**Abstract:** Dead wood is an important microhabitat for Uropodina mites (Acari: Mesostigmata). Earlier research has shown that dead wood contains about 1/3 of the Uropodina found so far in Poland, and its presence increases the overall biodiversity of forest ecosystems by 40%. The major aim of the current study is to assess the geographical variation of species diversity of Uropodina inhabiting dead wood in Poland and other European countries. The samples from dead wood (1180 samples in total) were collected in seven provinces in Poland, and in eight other countries (France, Italy, Belgium, The Netherlands, Slovakia, Sweden, Norway, and Denmark). Fifty-two Uropodina species were recovered from dead wood in seven provinces in Poland. The highest number of species was recorded in dead wood samples collected in Wielkopolskie, and the lowest in those from Zachodniopomorskie. The total number of species in the examined dead wood in the surveyed European countries was 24 species, ranging from 4 to 13 species per country. The most common species in the examined material from both Poland and other studied European countries were *Oodinychus ovalis* (C.L. Koch, 1839) and *Pulchellaobovella pulchella* (Berlese, 1904), though the frequency of the other species found in those areas was low. The differences in species diversity of the examined fauna of Uropodina probably result in the difference in the extent of the research carried out so far in some regions of Poland, lack of sufficient data for other European countries, as well as the highly diversified geographical ranges of most Uropodina species. The obtained results clearly show that there is still a need for further, more extensive research, based on a larger number of samples from dead wood from the whole continent.

**Keywords:** biodiversity; biogeography; merocenoses; Parasitiformes; unstable microhabitat; range of occurrence; saproxylic invertebrates; species richness

## 1. Introduction

Dead wood, along with bird and mammal nests, is one of the microhabitats most frequently colonized by mites from the suborder Uropodina (Acari: Mesostigmata) [1–8]. One of the precursors of the research into Uropodina inhabiting dead wood was Athias-Binche. This author conducted research in beech forests in France [1,2]. However, her research did not focus on the communities of these mites, but the effect of dead wood's decomposition degree on the number and population dynamic of one Uropodina species, i.e., *Allodinychus flagelliger* (Berlese, 1910).

For many years, several studies into Uropodina communities inhabiting various microhabitats, including dead wood, in different forests and in different areas of Poland has been carried out [4–10]. The results of this research show that almost 1/3 of all species known from Poland are found in dead wood [3,5–8]. So far, the diversity of Uropodina communities inhabiting different types of dead wood and the dependence of the species composition

of these mite communities on different tree species have been the major problems discussed in the published studies [6,8,10]. Thus, dead wood is an important element in increasing the biodiversity of forest ecosystems, both for the investigated mites and for other groups of organisms. The role of dead wood has been proved for saproxylic insects [11,12]. Among these, beetles are the best investigated so far as a group associated with this type of habitat [13–15]. Dead wood also constitutes habitats for other groups of invertebrates such as land snails and arachnids [16], as well as mites, including Oribatida [17,18]. Other groups of mites inhabiting dead wood include Mesostigmata, both predatory species and saprophagous or mycenophagous species feeding on mycelium and spores, and Uropodina are also among them [19–23]. As for the spatial differentiation of Uropodina occurring in dead wood, an analysis of the composition of these mite communities was carried out in three oak-hornbeam reserves in western Wielkopolska (Greater Poland). The research has revealed differences in the species composition of the communities inhabiting decayed wood and hollows at the level of one forest complex and in the whole country [10]. These observations, as well as the results of other studies on the biogeography of this group of mites, have shown that many species of the investigated mites have different range of occurrence in Poland [5,9,24]. This fact encouraged the authors of this study to analyze the species composition of Uropodina inhabiting decaying wood in Poland and other European countries.

The major aim of the current study was to compare the species composition of fauna of mites from suborder Uropodina inhabiting dead wood in different regions of Poland and to determine if and how the variation in geographical ranges of particular Uropodina species influences the species composition of Uropodina inhabiting decaying wood in different regions of the country. This article also discusses the state of the art in the research on Uropodina fauna occurring in microhabitats of dead wood in other European countries investigated so far, including France, Italy, Belgium, The Netherlands, Slovakia, Sweden, Norway, and Denmark.

## 2. Materials and Methods

### 2.1. Study Area

The material for this study comprises 1369 samples of dead wood (lying trunks, stumps, and hollows), collected in Poland (1180), France (25), Italy (20), Belgium (23), The Netherlands (20), Slovakia (27), Sweden (23), Norway (25), and Denmark (26) (Appendix A). The material collected in Poland comes from seven geographically different regions of the country (i.e., Zachodniopomorskie (28), Wielkopolskie (694), Kujawsko-Pomorskie (81), Podlaskie (147), Dolnośląskie (122), Podkarpackie (53), and Małopolskie (55)) (Figures 1 and 2).

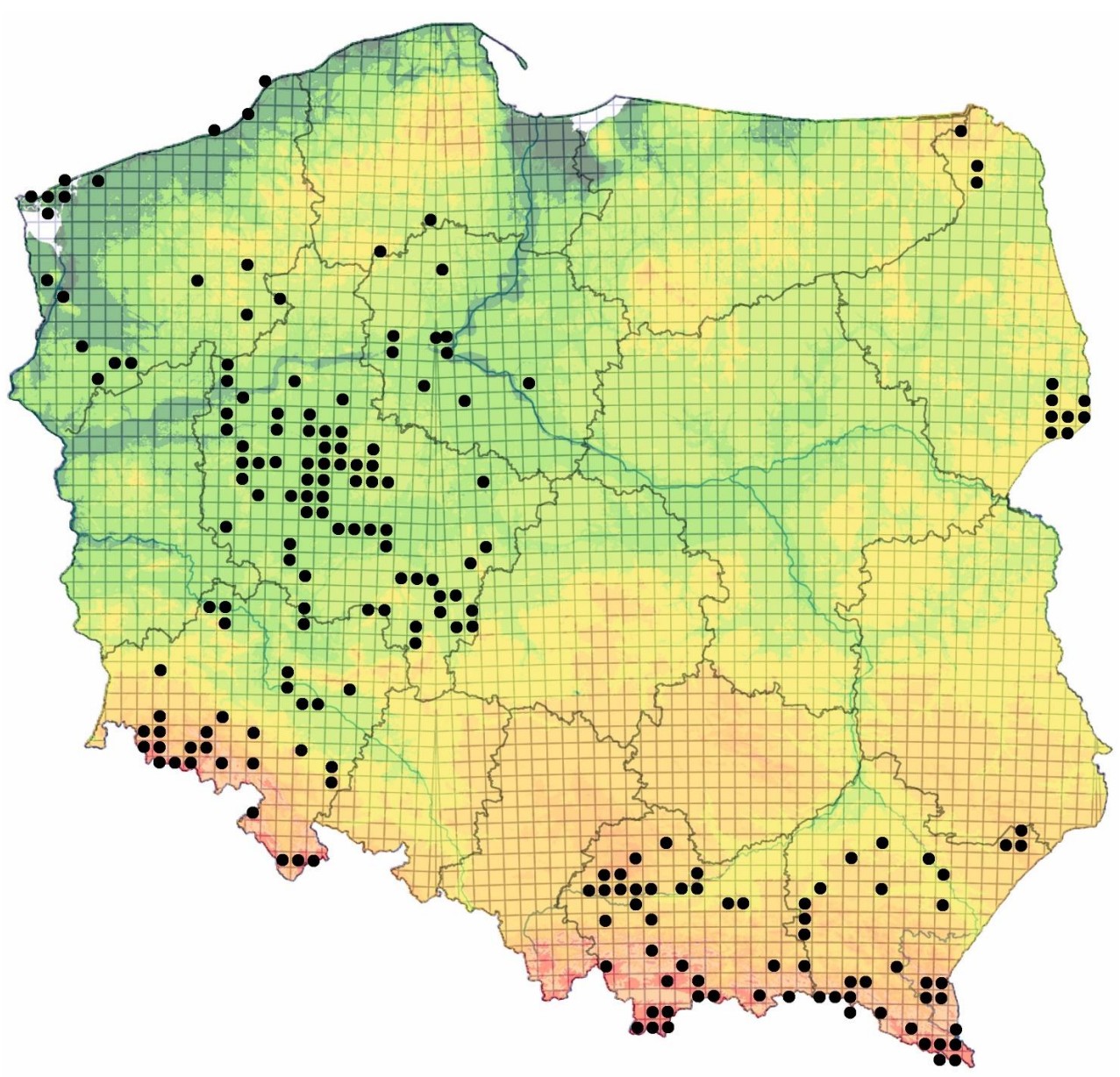

**Figure 1.** Location of the study sites in Poland (black dots) (UTM 10 × 10 km).

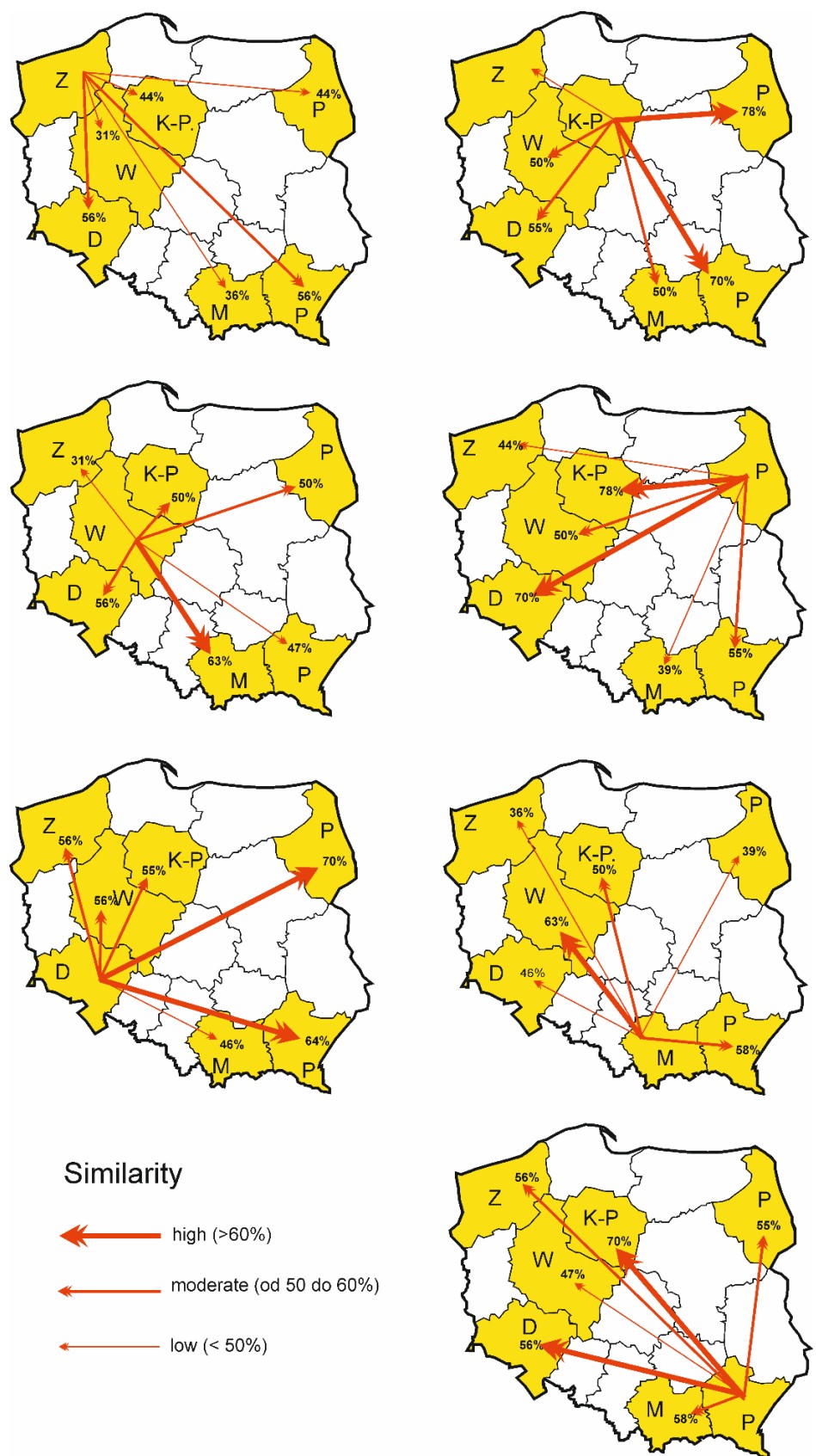

**Figure 2.** Species composition similarity (S) of Uropodina fauna in dead wood in examined provinces: Z—Zachodniopomorskie, K-P—Kujawsko-Pomorskie, W—Wielkopolskie, P—Podlaskie, D—Dolnośląskie, M—Małopolskie, Po—Podkarpackie.

*2.2. Data Collection*

All the material used in the analysis comprises samples from different species (Table A1) and types of decaying wood (including decaying stumps, logs, fallen branches, and hollows). The material from Poland and the other European countries was collected by the authors and co-workers using the same methods. The samples contained unsieved decaying wood with a volume between 0.5 and 0.8 L. The collected samples were extracted with Tullgren funnels for 4–6 days, depending on the level of material humidity. The extracted specimens of mites were then preserved in 75% ethanol. The mites were sorted out from the samples with a stereoscopic microscope and identified. Some species and juvenile stages were cleared in 80% lactic acid and identified by means of an Olympus BX51. The specimens were identified using the morphological criteria from the original descriptions and later accounts [3,5,25–27]. The identification of the species was carried out by the first author. All the collected specimens and samples have been deposited in the Invertebrate Fauna Bank (Natural History Collections, Faculty of Biology, Adam Mickiewicz University, Poznań, Poland).

*2.3. Data Analysis*

The zoocenological analysis of Uropodina fauna is based on the indices of dominance and frequency. The following classes were discerned [5]: Dominance: D5—eudominants (>30%), D4—dominants (15.1–30.0%), D3—subdominants (7.1–15.0%), D2—recedents (3.0–7.0%), and D1—subrecedents (<3%); Frequency: F5—euconstants (>50%), F4—constants (30.1–50%), F3—subconstants (15.1–30.0%), F2—accessory species (5.0–15.0%), and F1—accidents (<5%).

The similarity in the fauna of Uropodina found in decaying wood in the examined provinces was calculated by means of the Marczewski–Steinhaus species similarity index: MS = c/(a + b − c), where c is the number of species present in both compared communities, and a and b stand for the total numbers of species in each community [28]. The analyses were calculated with AnalizaTor 2.0 software (Poznań, Poland). Figure 1 was generated using MapaUTM ver. 5.4 [29]; Figure 2 was generated using CorelDRAW 2020 (18) ((64 Bit)—licence No. 382586, Poland, Poznań) (legal version).

**3. Results**

*3.1. Geographical Variation of Uropodina Fauna in Poland*

Fifty-two Uropodina species were found in the examined merocenoses of the seven Polish provinces (Table 1). Only seven (13.5%) species occurred in all the provinces. Nine species (17.3%) were found in the territory of 5 to 6 provinces, fourteen (26.9%) in the territory of 3 to 4 provinces, and the other species occurred merely from 1 to 2 provinces. The highest number of species was found in Wielkopolskie, and the lowest in Zachodniopomorskie.

Brief characteristics of the Uropodina fauna found in the examined provinces are presented below.

**Table 1.** Occurrence (+) of Uropodina mites in examined provinces: Z—Zachodniopomorskie, W—Wielkopolskie, K-P—Kujawsko-Pomorskie, P—Podlaskie, D—Dolnośląskie, Po—Podkarpackie, M—Małopolskie.

| Species | Z | W | K-P | P | D | Po | M |
|---|---|---|---|---|---|---|---|
| *Dinychus carinatus* Berlese, 1903 | + | + | + | + | + | + | + |
| *Pulchellaobovella pulchella* (Berlese, 1904) | + | + | + | + | + | + | + |
| *Uroobovella pyriformis* (Berlese, 1920) | + | + | + | + | + | + | + |
| *Olodiscus minima* (Kramer, 1882) | + | + | + | + | + | + | + |
| *Oodinychus ovalis* (C.L. Koch, 1839) | + | + | + | + | + | + | + |
| *Trachytes aegrota* (C.L. Koch, 1841) | + | + | + | + | + | + | + |
| *Trematurella elegans* (Kramer, 1882) | + | + | + | + | + | + | + |
| *Dinychus woelkiei* Hirschmann et Z.-Nicol, 1969 | + | + | + | + | + | + | |

**Table 1.** *Cont.*

| Species | Z | W | K-P | P | D | Po | M |
|---|---|---|---|---|---|---|---|
| *Trachytes pauperior* Berlese, 1914 | + | + |  | + | + | + | + |
| *Urodiaspis tecta* (Kramer, 1876) | + | + | + | + | + |  | + |
| *Dinychus arcuatus* (Trägårdh, 1943) |  | + | + | + | + | + | + |
| *Discourella baloghi* Hirschmann et Z.-Nicol, 1969 | + | + |  |  | + | + | + |
| *Polyaspis patavinus* Berlese, 1881 | + | + | + |  | + | + |  |
| *Pseudouropoda* sp. | + | + | + | + | + |  |  |
| *Dinychus perforatus* Kramer, 1882 |  | + | + | + | + | + | + |
| *Uroobovella* sp. |  | + |  | + | + | + | + |
| *Urodiaspis pannonica* Willmann, 1952 |  | + | + |  |  |  | + |
| *Iphiduropoda penicillata* (Hirschmann et Z.-Nicol, 1961) |  | + | + |  |  | + | + |
| *Neodiscopoma splendida* Kramer, 1882 |  | + | + |  |  | + | + |
| *Oodinychus karawaiewi* (Berlese, 1903) |  | + | + |  | + | + |  |
| *Uroobovella obovata* (Canestrini et Berlese, 1884) |  | + | + |  | + | + |  |
| *Olodiscus misella* (Berlese, 1916) |  |  | + |  | + | + | + |
| *Polyaspis sansonei* Berlese, 1916 | + | + |  | + |  |  |  |
| *Leiodinychus orbicularis* (C.L. Koch, 1839) |  | + | + | + |  |  |  |
| *Apionoseius infirmus* Berlese, 1887 |  | + |  | + | + |  |  |
| *Cilliba erlangensis* (Hirschmann et Z.-Nicol, 1969) |  | + |  |  |  |  | + |
| *Oodinychus obscurasimilis* (Hirschmann et Z.-Nicol, 1961) |  | + | + |  |  | + |  |
| *Phaulodiaspis rackei* (Oudemans, 1912) |  | + | + |  | + |  |  |
| *Polyaspinus cylindricus* Berlese, 1916 |  | + |  |  | + | + |  |
| *Trachytes irenae* Pecina, 1970 |  |  |  |  | + | + | + |
| *Uroobovella marginata* (C.L. Koch, 1839) | + | + |  |  |  |  |  |
| *Uropoda orbicularis* (Müller, 1776) | + |  |  |  |  | + |  |
| *Cilliba insularis* |  | + |  |  |  |  | + |
| *Dinychus inermis* (C.L. Koch, 1841) |  | + |  |  |  |  | + |
| *Discourella modesta* (Leonardi, 1889) |  | + |  |  | + |  |  |
| *Nenteria breviunguiculata* (Willmann, 1949) |  | + |  |  |  | + |  |
| *Nenteria stylifera* (Berlese, 1904) |  | + |  |  | + |  |  |
| *Pseudouropoda tuberosa* (Hirschmann et Z.-Nicol, 1961) |  | + |  |  | + |  |  |
| *Uroplitella paradoxa* (Canestrini et Berlese, 1884) |  | + |  |  |  |  | + |
| *Uropoda* sp. |  | + |  |  |  | + |  |
| *Trachytes minima* Trägårdh, 1910 |  |  |  |  |  | + | + |
| *Cilliba rafalskii* Błoszyk, Stachowiak, Halliday, 2006 |  | + |  |  |  |  |  |
| *Dinychura cordieri* (Berlese, 1916) |  | + |  |  |  |  |  |
| *Metagynella carpatica* (Balogh, 1943) |  | + |  |  |  |  |  |
| *Oplitis* sp. |  | + |  |  |  |  |  |
| *Pseudouropoda calcarata* (Hirschmann et Z.-Nicol, 1961) |  | + |  |  |  |  |  |
| *Trachyuropoda coccinea* (Michael, 1891) |  | + |  |  |  |  |  |
| *Trachytes lamda* Berlese, 1904 |  |  | + |  |  |  |  |
| *Oodinychus spatulifera* (Moniez, 1892) |  |  |  |  |  | + |  |
| *Polyaspinus schweizeri* (Hutu, 1976) |  |  |  |  |  | + |  |
| *Olodiscus kargi* (Hirschamann et Z.-Nicol, 1969) |  |  |  |  |  |  | + |
| *Trachytes montana* Willmann, 1953 |  |  |  |  |  |  | + |
| Number of species | 16 | 43 | 23 | 17 | 26 | 28 | 25 |
| Number of samples | 28 | 694 | 81 | 147 | 122 | 53 | 55 |

### 3.1.1. Zachodniopomorskie

In this province, 16 species of Uropodina were found in 28 samples from decaying wood (Table 2). The most dominant species, i.e., *O. ovalis* (42.66%) and *P. pulchella* (34.47%), constituted 77% of all the specimens of the Uropodina, and they were also the most frequent in the analyzed samples. This fauna was the most homogeneous of the surveyed provinces.

**Table 2.** Species composition of Uropodina fauna in dead wood in Zachodniopomorskie: N—number of specimens, D%—dominance, Avg. ± SD—average number in positive samples, F%—frequency, M—median, Max—maximum number of specimens in one sample.

| Species | N | D% | Avg. ± SD | F% | M | Max |
|---|---|---|---|---|---|---|
| *O. ovalis* | 250 | 42.66 | 27.78 ± 27.34 | 34.62 | 31.0 | 79 |
| *P. pulchella* | 202 | 34.47 | 33.67 ± 52.52 | 23.08 | 10.0 | 137 |
| *P. sansonei* | 42 | 7.17 | 42.00 | 3.85 | 42.0 | 42 |
| *P. patavinus* | 25 | 4.27 | 12.50 ± 16.26 | 7.69 | 12.5 | 24 |
| *U. pyriformis* | 20 | 3.41 | 6.67 ± 5.51 | 11.54 | 4.0 | 13 |
| *O. minima* | 14 | 2.39 | 3.50 ± 2.08 | 15.38 | 3.5 | 6 |
| *T. elegans* | 8 | 1.37 | 2.67 ± 2.08 | 11.54 | 2.0 | 5 |
| *Pseudouropoda* sp. | 6 | 1.02 | 3.00 | 7.69 | 3.0 | 3 |
| *D. baloghi* | 5 | 0.85 | 5.00 | 3.85 | 5.0 | 5 |
| *T. pauperior* | 4 | 0.68 | 4.00 | 3.85 | 4.0 | 4 |
| *U. marginata* | 3 | 0.51 | 1.50 ± 2.12 | 3.85 | 1.5 | 3 |
| *D. carinatus* | 2 | 0.34 | 1.00 | 7.69 | 1.0 | 1 |
| *D. woelkiei* | 2 | 0.34 | 2.00 | 3.85 | 2.0 | 2 |
| *T. aegrota* | 1 | 0.17 | 0.50 ± 0.71 | 3.85 | 0.5 | 1 |
| *U. tecta* | 1 | 0.17 | 1.00 | 3.85 | 1.0 | 1 |
| *U. orbicularis* | 1 | 0.17 | 1.00 | 3.85 | 1.0 | 1 |
| Total | 586 | 100 | | | | |

### 3.1.2. Wielkopolskie

The highest number of samples (694) from decaying wood was recorded in the material collected in Wielkopolskie. In these samples, 43 Uropodina species were found (Table 3). The most abundant species, such as in the case of Zachodniopomorskie, were *O. ovalis* (61.85%) and *P. pulchella* (14.18%), which constituted roughly 76% of the whole fauna. These species also had a high frequency in the samples. Common soil species such as *T. aegrota* and *O. minima* were also found frequently in the samples from the examined dead wood in Wielkopolskie

**Table 3.** Species composition of Uropodina fauna in dead wood in Wielkopolskie: N—number of specimens, D%—dominance, Avg. ± SD—average number in positive samples, F%—frequency, M—median, Max—maximum number of specimens in one sample.

| Species | N | D% | Avg. ± SD | F% | M | Max |
|---|---|---|---|---|---|---|
| *O. ovalis* | 9421 | 61.85 | 22.17 ± 36.21 | 60.09 | 8.0 | 236 |
| *P. pulchella* | 2160 | 14.18 | 15.88 ± 32.31 | 19.45 | 4.0 | 184 |
| *U. pyriformis* | 768 | 5.04 | 38.40 ± 72.53 | 2.59 | 11.5 | 315 |
| *O. minima* | 492 | 3.23 | 4.00 ± 5.28 | 16.71 | 2.0 | 32 |
| *T. aegrota* | 465 | 3.05 | 2.75 ± 3.27 | 23.20 | 1.0 | 18 |
| *D. baloghi* | 432 | 2.84 | 61.71 ± 159.75 | 1.01 | 1.0 | 424 |
| *D. carinatus* | 389 | 2.55 | 6.38 ± 13.83 | 8.21 | 2.0 | 105 |
| *D. woelkiei* | 240 | 1.58 | 5.71 ± 11.61 | 5.33 | 2.5 | 68 |
| *P. patavinus* | 231 | 1.52 | 28.88 ± 45.88 | 1.15 | 6.0 | 106 |
| *P. sansonei* | 104 | 0.68 | 9.46 ± 13.25 | 1.59 | 4.0 | 44 |
| *U. tecta* | 95 | 0.62 | 3.28 ± 3.93 | 4.18 | 2.0 | 14 |
| *O. karawaiewi* | 64 | 0.42 | 1.33 ± 4.41 | 2.31 | 0.0 | 30 |
| *L. orbicularis* | 57 | 0.37 | 4.75 ± 11.75 | 1.01 | 1.0 | 41 |
| *P. cylindricus* | 53 | 0.35 | 17.67 ± 21.22 | 0.43 | 8.0 | 42 |
| *A. infirmus* | 51 | 0.33 | 8.50 ± 11.26 | 0.86 | 1.5 | 24 |
| *T. pauperior* | 32 | 0.21 | 0.89 ± 0.95 | 2.88 | 1.0 | 3 |
| *D. perforatus* | 32 | 0.21 | 2.46 ± 3.48 | 1.30 | 1.0 | 11 |
| *U. obovata* | 30 | 0.20 | 3.00 ± 4.22 | 1.15 | 1.0 | 13 |
| *U. marginata* | 18 | 0.12 | 2.25 ± 6.36 | 0.14 | 0.0 | 18 |
| *I. penicillata* | 10 | 0.07 | 0.77 ± 1.48 | 0.58 | 0.0 | 5 |

**Table 3.** *Cont.*

| Species | N | D% | Avg. $\pm$ SD | F% | M | Max |
|---|---|---|---|---|---|---|
| *C. cassideasimilis* | 10 | 0.07 | 3.33 $\pm$ 2.08 | 0.43 | 4.0 | 5 |
| *Pseudouropoda* sp. | 9 | 0.06 | 0.82 $\pm$ 0.98 | 0.86 | 1.0 | 3 |
| *T. elegans* | 8 | 0.05 | 0.24 $\pm$ 0.61 | 0.86 | 0.0 | 3 |
| *D. cordieri* | 8 | 0.05 | 2.67 $\pm$ 2.89 | 0.43 | 1.0 | 6 |
| *D. arcuatus* | 8 | 0.05 | 0.80 $\pm$ 0.79 | 0.86 | 1.0 | 2 |
| *C. rafalskii* | 6 | 0.04 | 1.50 $\pm$ 1.29 | 0.43 | 1.5 | 3 |
| *T. coccinea* | 6 | 0.04 | 3.00 $\pm$ 1.41 | 0.29 | 3.0 | 4 |
| *Oplitis* sp. | 5 | 0.03 | 5.00 | 0.14 | 5.0 | 5 |
| *N. breviunguiculata* | 5 | 0.03 | 1.80 $\pm$ 0.45 | 0.72 | 2.0 | 2 |
| *Uroobovella* sp. | 4 | 0.03 | 2.00 $\pm$ 1.41 | 0.29 | 2.0 | 3 |
| *M. carpatica* | 3 | 0.02 | 1.50 $\pm$ 0.71 | 0.29 | 1.5 | 2 |
| *O. obscurasimilis* | 2 | 0.01 | 0.33 $\pm$ 0.82 | 0.14 | 0.0 | 2 |
| *P. calcarata* | 2 | 0.01 | 0.33 $\pm$ 0.82 | 0.14 | 0.0 | 2 |
| *U. pannonica* | 2 | 0.01 | 1.00 | 0.29 | 1.0 | 1 |
| *D. inermis* | 2 | 0.01 | 0.40 $\pm$ 0.89 | 0.14 | 0.0 | 2 |
| *D. modesta* | 1 | 0.01 | 1.00 | 0.14 | 1.0 | 1 |
| *P. tuberosa* | 1 | 0.01 | 0.17 $\pm$ 0.41 | 0.14 | 0.0 | 1 |
| *N. splendida* | 1 | 0.01 | 1.00 | 0.14 | 1.0 | 1 |
| *C. erlangensis* | 1 | 0.01 | 0.50 $\pm$ 0.71 | 0.14 | 0.5 | 1 |
| *P. rackei* | 1 | 0.01 | 1.00 | 0.14 | 1.0 | 1 |
| *U. paradoxa* | 1 | 0.01 | 1.00 | 0.14 | 1.0 | 1 |
| *Uropoda* sp. | 1 | 0.01 | 1.00 | 0.14 | 1.0 | 1 |
| *N. stylifera* | 1 | 0.01 | 1.00 | 0.14 | 1.0 | 1 |
| Total | 15,232 | 100 | | | | |

### 3.1.3. Kujawsko-Pomorskie

In the 81 samples collected in Kujawsko-Pomorskie, 23 Uropodina species were found (Table 4). The most numerous and most common species in the examined fauna were *P. pulchella* (38.10%) and *O. ovalis* (33.80%). They constituted 72% of the total number of all specimens in the fauna.

**Table 4.** Species composition of Uropodina fauna in dead wood in Kujawsko-Pomorskie: N—number of specimens, D%—dominance, Avg. $\pm$ SD—average number in positive samples, F%—frequency, M—median, Max—maximum number of specimens in one sample.

| Species | N | D% | Avg. $\pm$ SD | F% | M | Max |
|---|---|---|---|---|---|---|
| *P. pulchella* | 328 | 38.10 | 32.80 $\pm$ 41.61 | 12.35 | 14.0 | 113 |
| *O. ovalis* | 291 | 33.80 | 8.31 $\pm$ 7.82 | 43.21 | 6.0 | 27 |
| *D. perforatus* | 64 | 7.43 | 10.67 $\pm$ 14.01 | 7.41 | 2.5 | 32 |
| *D. arcuatus* | 28 | 3.25 | 14.00 $\pm$ 12.73 | 2.47 | 14.0 | 23 |
| *T. elegans* | 21 | 2.44 | 3.00 $\pm$ 2.45 | 8.64 | 2.0 | 8 |
| *O. obscurasimilis* | 19 | 2.21 | 4.75 $\pm$ 6.85 | 4.94 | 1.5 | 15 |
| *T. aegrota* | 15 | 1.74 | 1.67 $\pm$ 0.87 | 11.11 | 1.0 | 3 |
| *U. tecta* | 15 | 1.74 | 3.00 $\pm$ 3.46 | 6.17 | 1.0 | 9 |
| *O. minima* | 9 | 1.05 | 1.29 $\pm$ 0.49 | 8.64 | 1.0 | 2 |
| *N. splendida* | 9 | 1.05 | 9.00 | 1.23 | 9.0 | 9 |
| *L. orbicularis* | 8 | 0.93 | 4.00 $\pm$ 4.24 | 2.47 | 4.0 | 7 |
| *U. pannonica* | 8 | 0.93 | 2.00 $\pm$ 2.00 | 4.94 | 1.0 | 5 |
| *D. carinatus* | 8 | 0.93 | 2.67 $\pm$ 2.08 | 3.70 | 2.0 | 5 |
| *T. lamda* | 7 | 0.81 | 7.00 | 1.23 | 7.0 | 7 |
| *O. karawaiewi* | 7 | 0.81 | 3.50 $\pm$ 2.12 | 2.47 | 3.5 | 5 |
| *D. woelkiei* | 6 | 0.70 | 2.00 $\pm$ 1.73 | 3.70 | 1.0 | 4 |
| *I. penicillata* | 5 | 0.58 | 2.50 $\pm$ 2.12 | 2.47 | 2.5 | 4 |
| *U. pyriformis* | 5 | 0.58 | 5.00 | 1.23 | 5.0 | 5 |
| *O. misella* | 2 | 0.23 | 2.00 | 1.23 | 2.0 | 2 |

**Table 4.** *Cont.*

| Species | N | D% | Avg. $\pm$ SD | F% | M | Max |
|---|---|---|---|---|---|---|
| *P. patavinus* | 1 | 0.12 | 1.00 | 1.23 | 1.0 | 1 |
| *Pseudouropoda* sp. | 1 | 0.12 | 1.00 | 1.23 | 1.0 | 1 |
| *U. obovata* | 1 | 0.12 | 1.00 | 1.23 | 1.0 | 1 |
| *P. rackei* | 1 | 0.12 | 1.00 | 1.23 | 1.0 | 1 |
| Total | 859 | 100 | | | | |

3.1.4. Podlaskie

A total of 147 samples were collected in Podlaskie, where 17 Uropodina species were found (Table 5). *Oodinychus ovalis* (49.97%) and *D. carinatus* (19.82%) were the two most dominant species, which constituted 70% of the whole fauna. *Pulchellaobovella pulchella* was also quite abundant and frequent, just like *T. aegrota* and *D. arcuatus*.

**Table 5.** Species composition of Uropodina fauna in dead wood in Podlaskie: N—number of specimens, D%—dominance, Avg. $\pm$ SD—average number in positive samples, F%—frequency, M—median, Max—maximum number of specimens in one sample.

| Species | N | D% | Avg. $\pm$ SD | F% | M | Max |
|---|---|---|---|---|---|---|
| *O. ovalis* | 928 | 49.97 | 7.55 $\pm$ 17.78 | 51.02 | 2.0 | 134 |
| *D. carinatus* | 368 | 19.82 | 4.04 $\pm$ 21.69 | 9.52 | 0.0 | 164 |
| *P. pulchella* | 166 | 8.94 | 2.24 $\pm$ 10.23 | 12.93 | 0.0 | 86 |
| *T. aegrota* | 144 | 7.75 | 1.26 $\pm$ 3.77 | 21.77 | 0.0 | 28 |
| *D. arcuatus* | 111 | 5.98 | 1.26 $\pm$ 3.94 | 11.56 | 0.0 | 23 |
| *D. woelkiei* | 43 | 2.32 | 0.54 $\pm$ 3.83 | 3.40 | 0.0 | 34 |
| *Pseudouropoda* sp. | 23 | 1.24 | 0.26 $\pm$ 1.14 | 4.76 | 0.0 | 8 |
| *U. pyriformis* | 20 | 1.08 | 0.46 $\pm$ 2.87 | 1.36 | 0.0 | 19 |
| *T. elegans* | 14 | 0.75 | 0.14 $\pm$ 0.45 | 7.48 | 0.0 | 3 |
| *U. tecta* | 11 | 0.59 | 0.23 $\pm$ 0.72 | 4.08 | 0.0 | 4 |
| *T. pauperior* | 8 | 0.43 | 0.10 $\pm$ 0.51 | 2.72 | 0.0 | 4 |
| *A. infirmus* | 4 | 0.22 | 4.00 | 0.68 | 4.0 | 4 |
| *P. sansonei* | 4 | 0.22 | 0.06 $\pm$ 0.30 | 2.04 | 0.0 | 2 |
| *Uroobovella* sp. | 4 | 0.22 | 0.06 $\pm$ 0.23 | 2.72 | 0.0 | 1 |
| *O. minima* | 4 | 0.22 | 0.11 $\pm$ 0.52 | 1.36 | 0.0 | 3 |
| *D. perforatus* | 4 | 0.22 | 0.05 $\pm$ 0.22 | 2.72 | 0.0 | 1 |
| *L. orbicularis* | 1 | 0.05 | 1.00 | 0.68 | 1.0 | 1 |
| Total | 1857 | 100 | | | | |

3.1.5. Dolnośląskie

In 122 samples from Dolnośląskie, 26 Uropodina species were found (Table 6). The most abundant species was *O. ovalis*, which constituted 63% of the whole fauna. The next most abundant species were *T. aegrota* and *U. pyriformis*. The frequency in the samples of the most dominant species was relatively low (less than 20%).

**Table 6.** Species composition of Uropodina fauna in dead wood in Dolnośląskie: N—number of specimens, D%—dominance, Avg. $\pm$ SD—average number in positive samples, F%—frequency, M—median, Max—maximum number of specimens in one sample.

| Species | N | D% | Avg. $\pm$ SD | F% | M | Max |
|---|---|---|---|---|---|---|
| *O. ovalis* | 813 | 63.12 | 35.35 $\pm$ 53.41 | 18.85 | 12.0 | 216 |
| *T. aegrota* | 76 | 5.90 | 5.85 $\pm$ 12.24 | 10.66 | 1.0 | 46 |
| *U. pyriformis* | 72 | 5.59 | 24.00 $\pm$ 28.36 | 2.46 | 14.0 | 56 |
| *D. woelkiei* | 54 | 4.19 | 27.00 $\pm$ 36.77 | 1.64 | 27.0 | 53 |
| *P. pulchella* | 47 | 3.65 | 4.70 $\pm$ 6.20 | 8.20 | 2.0 | 19 |
| *U. tecta* | 33 | 2.56 | 4.13 $\pm$ 6.56 | 6.56 | 1.5 | 20 |

**Table 6.** *Cont.*

| Species | N | D% | Avg. $\pm$ SD | F% | M | Max |
|---|---|---|---|---|---|---|
| *O. karawaiewi* | 29 | 2.25 | 9.67 $\pm$ 7.77 | 2.46 | 12.0 | 16 |
| *O. minima* | 28 | 2.17 | 4.00 $\pm$ 4.00 | 5.74 | 2.0 | 11 |
| *Pseudouropoda* sp. | 20 | 1.55 | 10.00 $\pm$ 8.49 | 1.64 | 10.0 | 16 |
| *P. cylindricus* | 18 | 1.40 | 9.00 $\pm$ 9.90 | 1.64 | 9.0 | 16 |
| *P. patavinus* | 17 | 1.32 | 8.50 $\pm$ 2.12 | 1.64 | 8.5 | 10 |
| *D. arcuatus* | 16 | 1.24 | 16.00 | 0.82 | 16.0 | 16 |
| *T. pauperior* | 14 | 1.09 | 2.00 $\pm$ 1.83 | 5.74 | 1.0 | 6 |
| *P. tuberosa* | 11 | 0.85 | 5.50 $\pm$ 0.71 | 1.64 | 5.5 | 6 |
| *N. stylifera* | 11 | 0.85 | 5.50 $\pm$ 6.36 | 1.64 | 5.5 | 10 |
| *D. carinatus* | 7 | 0.54 | 3.50 $\pm$ 3.54 | 1.64 | 3.5 | 6 |
| *D. baloghi* | 4 | 0.31 | 2.00 $\pm$ 1.41 | 1.64 | 2.0 | 3 |
| *A. infirmus* | 3 | 0.23 | 1.50 $\pm$ 0.71 | 1.64 | 1.5 | 2 |
| *T. elegans* | 3 | 0.23 | 3.00 | 0.82 | 3.0 | 3 |
| *O. misella* | 3 | 0.23 | 1.50 $\pm$ 0.71 | 1.64 | 1.5 | 2 |
| *D. perforatus* | 3 | 0.23 | 3.00 | 0.82 | 3.0 | 3 |
| *D. modesta* | 2 | 0.16 | 2.00 | 0.82 | 2.0 | 2 |
| *T. irenae* | 1 | 0.08 | 1.00 | 0.82 | 1.0 | 1 |
| *U. obovata* | 1 | 0.08 | 1.00 | 0.82 | 1.0 | 1 |
| *Uroobovella* sp. | 1 | 0.08 | 1.00 | 0.82 | 1.0 | 1 |
| *P. rackei* | 1 | 0.08 | 1.00 | 0.82 | 1.0 | 1 |
| Total | 1288 | 100 | | | | |

### 3.1.6. Małopolskie

In 55 samples collected in Małopolskie, 25 Uropodina species were found (Table 7). The most numerous specimens of the discussed group of mites were *T. aegrota* (37.69%) and *O. ovalis* (19.15%), which constituted 57% of the fauna. Their frequency in the samples was low (below 25%). In the analyzed fauna, a relatively high percentage of *T. irenae*, which is the third most abundant species, was recorded.

**Table 7.** Species composition of Uropodina fauna in dead wood in Małopolskie: N—number of specimens, D%—dominance, Avg. $\pm$ SD—average number in positive samples, F%—frequency, M—median, Max—maximum number of specimens in one sample.

| Species | N | D% | Avg. $\pm$ SD | F% | M | Max |
|---|---|---|---|---|---|---|
| *T. aegrota* | 248 | 37.69 | 17.71 $\pm$ 40.53 | 23.64 | 3.0 | 150 |
| *O. ovalis* | 126 | 19.15 | 14.00 $\pm$ 15.12 | 16.36 | 6.0 | 44 |
| *T. irenae* | 48 | 7.29 | 8.00 $\pm$ 5.87 | 10.91 | 8.5 | 15 |
| *D. baloghi* | 47 | 7.14 | 9.40 $\pm$ 17.67 | 9.09 | 2.0 | 41 |
| *D. carinatus* | 38 | 5.78 | 5.43 $\pm$ 6.08 | 12.73 | 3.0 | 18 |
| *D. arcuatus* | 28 | 4.26 | 9.33 $\pm$ 1.53 | 5.45 | 9.0 | 11 |
| *T. minima* | 17 | 2.58 | 8.50 $\pm$ 7.78 | 3.64 | 8.5 | 14 |
| *U. tecta* | 16 | 2.43 | 3.20 $\pm$ 4.92 | 9.09 | 1.0 | 12 |
| *C. cassideasimilis* | 16 | 2.43 | 8.00 $\pm$ 9.90 | 3.64 | 8.0 | 15 |
| *N. splendida* | 12 | 1.82 | 4.00 $\pm$ 4.36 | 5.45 | 2.0 | 9 |
| *U. pyriformis* | 10 | 1.52 | 2.00 $\pm$ 0.71 | 9.09 | 2.0 | 3 |
| *D. perforatus* | 9 | 1.37 | 9.00 | 1.82 | 9.0 | 9 |
| *T. pauperior* | 6 | 0.91 | 2.00 $\pm$ 1.73 | 5.45 | 1.0 | 4 |
| *P. pulchella* | 6 | 0.91 | 2.00 $\pm$ 1.00 | 5.45 | 2.0 | 3 |
| *O. misella* | 6 | 0.91 | 6.00 | 1.82 | 6.0 | 6 |
| *U. pannonica* | 5 | 0.76 | 2.50 $\pm$ 2.12 | 3.64 | 2.5 | 4 |
| *O. minima* | 5 | 0.76 | 2.50 $\pm$ 0.71 | 3.64 | 2.5 | 3 |
| *Uroobovella* sp. | 3 | 0.46 | 1.00 | 5.45 | 1.0 | 1 |
| *O. kargi* | 3 | 0.46 | 3.00 | 1.82 | 3.0 | 3 |
| *C. erlangensis* | 3 | 0.46 | 1.50 $\pm$ 0.71 | 3.64 | 1.5 | 2 |
| *T. elegans* | 2 | 0.30 | 2.00 | 1.82 | 2.0 | 2 |

**Table 7.** *Cont.*

| Species | N | D% | Avg. $\pm$ SD | F% | M | Max |
|---|---|---|---|---|---|---|
| *T. montana* | 1 | 0.15 | 1.00 | 1.82 | 1.0 | 1 |
| *I. penicillata* | 1 | 0.15 | 1.00 | 1.82 | 1.0 | 1 |
| *U. paradoxa* | 1 | 0.15 | 1.00 | 1.82 | 1.0 | 1 |
| *D. inermis* | 1 | 0.15 | 1.00 | 1.82 | 1.0 | 1 |
| Total | 658 | 100 | | | | |

### 3.1.7. Podkarpackie

As many as 28 species of Uropodina were found in 53 samples from the Podkarpackie (Table 8). The most abundant species in the fauna were *O. ovalis* (31.75%) and *D. baloghi* (19.26%), which constitute 51% of all specimens found there. However, their frequency in the samples was not high (less than 25%).

**Table 8.** Species composition of Uropodina fauna in dead wood in Podkarpackie: N—number of specimens, D%—dominance, Avg. $\pm$ SD—average number in positive samples, F%—frequency, M—median, Max—maximum number of specimens in one sample.

| Species | N | D% | Avg. $\pm$ SD | F% | M | Max |
|---|---|---|---|---|---|---|
| *O. ovalis* | 181 | 31.75 | 13.92 $\pm$ 14.22 | 24.53 | 5.0 | 39 |
| *D. baloghi* | 109 | 19.26 | 27.25 $\pm$ 30.10 | 7.55 | 21.0 | 66 |
| *D. woelkiei* | 53 | 9.36 | 17.67 $\pm$ 26.27 | 5.66 | 3.0 | 48 |
| *P. pulchella* | 37 | 6.54 | 5.29 $\pm$ 3.25 | 13.21 | 4.0 | 10 |
| *N. splendida* | 31 | 5.48 | 7.75 $\pm$ 12.84 | 7.55 | 1.5 | 27 |
| *D. arcuatus* | 28 | 4.95 | 7.00 $\pm$ 11.34 | 7.55 | 1.5 | 24 |
| *D. carinatus* | 27 | 4.77 | 13.50 $\pm$ 14.85 | 3.77 | 13.5 | 24 |
| *T. aegrota* | 16 | 2.83 | 1.78 $\pm$ 1.99 | 13.21 | 1.0 | 6 |
| *O. misella* | 15 | 2.65 | 7.50 $\pm$ 9.19 | 3.77 | 7.5 | 14 |
| *O. spatulifera* | 14 | 2.47 | 3.50 $\pm$ 3.79 | 7.55 | 2.0 | 9 |
| *U. pyriformis* | 11 | 1.94 | 5.50 $\pm$ 4.95 | 3.77 | 5.5 | 9 |
| *Uropoda* sp. | 9 | 1.59 | 4.50 $\pm$ 4.95 | 3.77 | 4.5 | 8 |
| *T. pauperior* | 5 | 0.88 | 1.67 $\pm$ 0.58 | 5.66 | 2.0 | 2 |
| *P. cylindricus* | 4 | 0.71 | 1.33 $\pm$ 0.58 | 5.66 | 1.0 | 2 |
| *P. patavinus* | 4 | 0.71 | 4.00 | 1.89 | 4.0 | 4 |
| *D. perforatus* | 4 | 0.71 | 2.00 $\pm$ 1.41 | 3.77 | 2.0 | 3 |
| *T. minima* | 3 | 0.53 | 1.50 $\pm$ 0.71 | 3.77 | 1.5 | 2 |
| *U. obovata* | 3 | 0.53 | 3.00 | 1.89 | 3.0 | 3 |
| *T elegans* | 2 | 0.35 | 2.00 | 1.89 | 2.0 | 2 |
| *Uroobovella* sp. | 2 | 0.35 | 2.00 | 1.89 | 2.0 | 2 |
| *T. irenae* | 1 | 0.18 | 1.00 | 1.89 | 1.0 | 1 |
| *P. schweizeri* | 1 | 0.18 | 1.00 | 1.89 | 1.0 | 1 |
| *O. karawaiewi* | 1 | 0.18 | 1.00 | 1.89 | 1.0 | 1 |
| *O. obscurasimilis* | 1 | 0.18 | 1.00 | 1.89 | 1.0 | 1 |
| *I. penicillata* | 1 | 0.18 | 1.00 | 1.89 | 1.0 | 1 |
| *O. minima* | 1 | 0.18 | 1.00 | 1.89 | 1.0 | 1 |
| *U. orbicularis* | 1 | 0.18 | 1.00 | 1.89 | 1.0 | 1 |
| *N. breviunguiculata* | 1 | 0.18 | 1.00 | 1.89 | 1.0 | 1 |
| Total | 566 | 100 | | | | |

### 3.2. Geographical Similarity of Species Composition of Examined Fauna

The geographical distribution of the examined provinces allowed for an analysis of the similarity of the species composition of Uropodina in different regions of Poland. Three major classes of this similarity have been discerned:

I—low (<50%);

II—moderate (from 50 to 60%);

III—high (>60%).

Figures 2 and 3 show the percentage value of species similarity of Uropodina fauna inhabiting dead wood in the examined provinces. The Marczewski–Steinhaus species similarity index of Uropodina inhabiting decaying wood in the examined provinces showed a great difference in the species composition in Małopolskie and Wielkopolskie (Figure 3). The fauna in Kujawsko-Pomorskie and Podlaskie, where the samples were collected largely from two sites of high natural value (such as Białowieża Forest (in Podlaskie) and Cisy Staropolskie im. L. Wyczółkowski Reserve in Wierzchlas (in Kujawsko-Pomorskie), turned out to be the most similar.

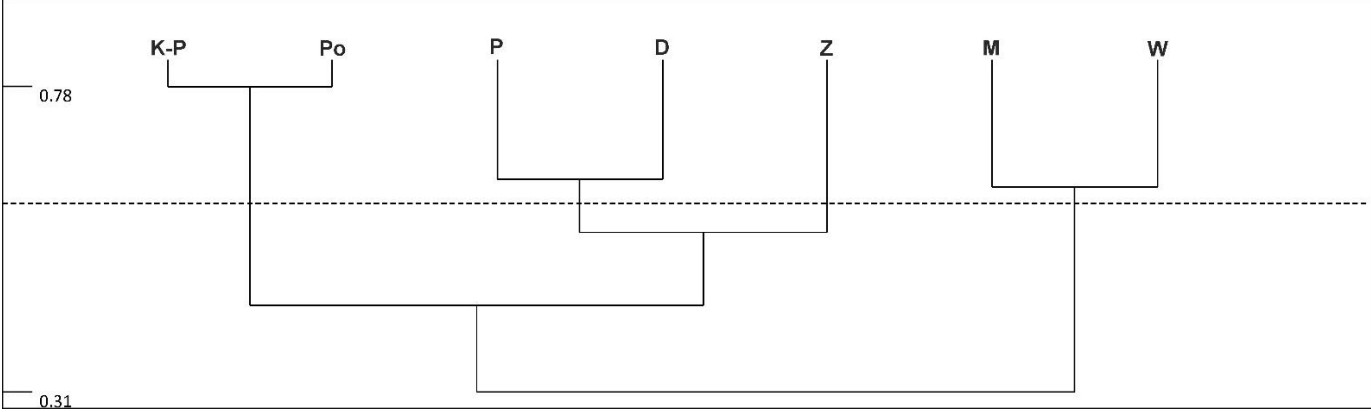

**Figure 3.** Species composition similarity (S) of Uropodina inhabiting dead wood in examined provinces: Z—Zachodniopomorskie, K-P—Kujawsko-Pomorskie, W—Wielkopolskie, P—Podlaskie, D—Dolnośląskie, M—Małopolskiea, Po—Podkarpackie.

The fauna of Uropodina in Zachodniopomorskie is characterized by a low similarity among the other examined provinces (Figures 2 and 3). However, it is fairly similar to Dolnośląskie and Podkarpackie (Figure 3). In Wielkopolskie, where the highest number of Uropodina species was found, the fauna is the least similar to the north-western areas of Poland (Zachodniopomorskie) and Podkarpackie, which is the most south-eastern province. Furthermore, the fauna examined in Kujawsko-Pomorskie is characterized by moderate similarity to the fauna in Wielkopolskie, Dolnośląskie, and Małopolskie, and by high similarity to the fauna in Podlaskie and Podkarpackie. The fauna of Uropodina in Podlaskie was least similar to those in Zachodniopomorskie and Małopolskie, and most similar to those in Kujawsko-Pomorskie and Dolnośląskie. The examined fauna in Dolnośląskie was least similar to the fauna in Małopolskie, and very similar to the fauna in Podlaskie and Podkarpackie. The fauna in Małopolskie was characterized by moderate similarity of species composition to the fauna found in Kujawsko-Pomorskie and Podkarpackie, and they exhibited a high similarity to the fauna examined in Wielkopolskie. In Podkarpackie, which is the most south-eastern province in Poland, the analyzed fauna was characterized by the highest species similarity of Uropodina to those found in Kujawsko-Pomorskie and Dolnośląskie.

It is worth mentioning that the material from dead wood was collected from 33 species of trees (Table A1). In Wielkopolskie, which was the most thoroughly examined province, the dead wood samples came from 29 species of trees (Table A1), and in the other provinces, the number of tree species ranged from 8 to 15. Most of the material originated from six tree species (oak, beech, horse-chestnut, linden, pine, and spruce). Three tree species (oak, beech, and horse-chestnut) were represented in the material from all the examined provinces, whereas in three provinces, dead wood was not collected from pine (Podkarpackie), linden (Podlaskie), and spruce (Kujawsko-Pomorskie). The percentage of the decaying wood from particular tree species in the analyzed material varied in the provinces, which was probably one of the factors differentiating the species composition of the found mites.

### 3.3. Species Diversity of Uropodina in Dead Wood in Selected European Countries

Samples of dead wood collected in several European countries stored in the Natural History Collections allowed to establish the species diversity of Uropodina mites (Table 9). Obviously, the results presented here are only preliminary due to the insufficient number of samples. However, as the number of samples was relatively similar, these results should be considered reliable.

**Table 9.** Uropodina species found in dead wood samples collected in some European countries: Fr—France, It—Italy, Be—Belgium, Ne—The Netherlands, Sl—Slovakia, De—Denmark, Sw—Sweden, No—Norway, N—number.

| | Fr | It | Be | Ne | Sl | De | Sw | No |
|---|---|---|---|---|---|---|---|---|
| N of samples | 25 | 20 | 23 | 20 | 27 | 26 | 23 | 25 |
| N of specimens | 15 | 148 | 183 | 197 | 70 | 283 | 154 | 29 |
| N of species | 5 | 7 | 4 | 5 | 4 | 13 | 4 | 7 |
| Species | | | | | | | | |
| *O. ovalis* | + | + | + | + | | + | + | |
| *O. minima* | + | | | + | | + | + | |
| *P. pulchella* | | | + | + | | + | | + |
| *D. woelkei* | | | + | | | + | | + |
| *T. aegrota* | + | | | + | | + | | |
| *C. cassidea* | + | | | | | | + | |
| *D. arcuatus* | | | | | | + | | + |
| *D. carinatus* | | | + | | | + | | |
| *D. perforatus* | | | | | + | | | + |
| *P. patavinus* | | + | | | + | | | |
| *Pseudouropoda* sp. | | + | | | | + | | |
| *U. pyriformis* | | + | | | + | | | |
| *D. cordieri* | | | | | | | + | |
| *Metagynella* sp. | | | | | | + | | |
| *Microurobovella olszanowski* | | + | | | | | | |
| *O. karawaiewi* | | | | | + | | | |
| *O. misella* | | | | | | + | | |
| *Oplitis* sp. | | + | | | | | | |
| *P. cylindricus* | + | | | | | | | |
| *Trachytes* sp. | | + | | | | | | |
| *U. flageliger* | | | | | | + | | |
| *Uroobovella* sp. | | | | | | + | | |
| *U. obovata* | | | | | | + | | |
| *U. tecta* | | | | | + | | | |

The total number of species in dead wood in eight investigated European countries (except for Poland) was 24 species, ranging from 4 to 13 species per country. The lowest number of species in the dead wood samples was revealed in the samples from Belgium, Slovakia, and Sweden. The highest number of species was recorded in the samples from Denmark. The species that occurred most frequently in six out of the eight countries was *O. ovalis*. Moreover, *O. minima* and *P. pulchella* occurred in the samples from 1/2 of the examined countries, while *D. woelkei* and *T. aegrota* occurred in three countries. The other 19 species occurred only once or twice in the analyzed material from the investigated countries.

## 4. Discussion

The fauna of Uropodina is highly diverse in decaying wood throughout Poland. The highest number of species (43) (Table 1) was found in Wielkopolskie, whereas the lowest (16 each) in Zachodniopomorskie and Podlaskie. In the other provinces, the number of species was similar (ranging between 23 and 28). Such a large difference in the number

of species between Wielkopolskie, Zachodniopomorskie, and Podlaskie certainly results from the differences in the extent of the research in these regions, i.e., a considerably higher number of samples collected in Wielkopolskie. It is also worth noting that over the course of its climatic history, the northern areas of Poland were covered by Pleistocene glaciations as many as four times [30,31]. Therefore, the species diversity of Uropodina may be the lowest here due to the shortest time the mites had to colonize these areas. At this stage, however, it is difficult to assess which of the aforementioned facts, whether it is the extent of the research or natural conditions resulting from the geological history of Poland and the rate of species dispersal, is responsible for this situation. That is why further research based on a larger number of samples from various microhabitats from different parts of Poland is necessary.

The fauna of Uropodina inhabiting dead wood is specific, not only due to their species composition, but also their community structure. Numerous uropodine mites are found only in decaying wood, resulting in a considerably different species composition within this microhabitat in contrast to the surrounding soil and leaf litter [6,10]. Community structure within dead wood is also different from those of soil and leaf litter, with usually just two most abundant species which constitute roughly between 51 and 77% of the whole community [6]. A similar dominance structure of Uropodina in dead wood was found in our study. *Oodinychus ovalis* was the most or one of the most dominant species in all seven provinces (from 19.15% up to 63.12%). The second species which was one of the dominant species in three provinces (from 38.10% to 14.18%) is *P. pulchella*. A different structure of dominance was observed in the fauna found in Małopolskie and Dolnośląskie, where one of the dominant species was *T. aegrota*, while in Podlaskie, the most numerous species was *D. carinatus*, and in Podkarpackie, *D. baloghi*. An earlier study shows that Uropodina inhabiting decaying wood are characterized by their low frequency in samples [3], which probably reduces species diversity in our results. Thus, the large number of samples collected in the Wielkopolskie is directly related to the high number of species found in the samples of dead wood from this area. In the analyzed material, only *O. ovalis*, which is one of the most numerous and common Uropodina species in Poland [5], had a frequency higher than 50% in two provinces (Wielkopolskie and Podlaskie). The majority of Uropodina found in this microhabitat are sub-constants, accessory, or even accidental species, whose frequency is lower than 5%.

Patterns of fauna diversity could also be caused by other factors not considered in this study, such as tree species, humidity, and the degree of wood decay. The tree species of the dead wood in which the species diversity of Uropodina mites had the highest value are pine, oak, beech, and hornbeam [8]. Uropodina mites prefer highly decomposed wood, which is moderately humid. In contrast, they avoid extremely dry and very wet areas [10]. The type of the dead wood is also important. These factors are crucial because species composition and abundance of the Uropodina differ in lying, decaying logs, stumps of felled trees, root-rotted live trees, and in hollows [4,10]. Thus, to determine the actual species diversity of Uropodina in decaying wood, it is necessary to collect a very large number of samples from each province, taking into account as many tree species as possible, all types of dead wood, different types of forest ecosystems, and the degree of wood decay.

The variation in the species diversity of Uropodina inhabiting dead wood in different regions of Poland, similarly to mite communities inhabiting soil [24], is also influenced by the variation in the geographical ranges of individual species. The presented studies show that such widely distributed species as *P. pulchella*, *U. pyriformis*, *O. minima*, *O. ovalis*, *T. aegrota*, *D. carinatus*, and *T. elegans* are usually the most abundant and common species in mite communities in dead wood [5,6,8,10,24]. It is also likely that species such as *D. woelkiei*, *T. pauperior*, *U. tecta*, *D. arcuatus*, *D. baloghi*, *P. patavinus*, and *D. perforatus* are widely distributed and can be found in decaying wood in all 16 provinces of Poland. On the other hand, species that do not occur in the whole area of Poland but have varied ranges in this area [5,24] were found in the dead wood samples from selected provinces. Among such species are *T. irenae*, *T. montana*, and *T. minima*, which have their northern limits in

Poland [24] and were found only in the samples from the southern provinces (Dolnośląskie, Podkarpackie, Małopolskie). The group of species with different ranges of occurrence also includes *Polyaspinus schweizeri*, which is an east-Carpathian species [24], occurring only in the material from Podkarpackie. Furthermore, species such as *C. erlangensis*, *O. misella,* and *P. cylindricus*, whose range in Poland overlaps with that of the beech (*Fagus sylvaticus*) [24], occurred in Dolnośląskie, Podkarpackie, Małopolskie, Wielkopolskie, and in Kujawsko-Pomorskie. *N. splendida*, which has a disjunct range in Poland [24], was found in the dead wood samples from Wielkopolskie and Kujawsko-Pomorskie, and in the south, including the samples from Podkarpackie and Małopolskie.

In general, the low species diversity of Uropodina fauna in decaying wood from the other European countries, which differs considerably from that observed in Poland, is due to the small number of samples from these countries. However, already at this preliminary stage of the study, it can be concluded that the dominance structure in Uropodina fauna in dead wood in those countries is probably similar to that observed in Poland. In the material collected from the investigated European countries, the most common species were also *O. ovalis*, *P. pulchella,* and *O. minima* (Table 9). *Oodinychus ovalis* (Figure 4) and *O. minima* are widely distributed species throughout Europe, but *P. pulchella* had, according to previous studies, a range restricted to central and southern Europe [5], but the observations presented in this study show that it also occurs in Denmark, Belgium, The Netherlands, and Norway. Further research could confirm the occurrence in microhabitats of dead wood in different European countries of a similar, or even higher, number of Uropodina species, different from those found so far in Poland. A good example of such a species, associated with dead wood in other areas of Europe, is *M. olszanowski*, which has been found in Italy [32] and should be considered as endemic for the time being. Another example is the occurrence of numerous *A. flagelliger* in the dead wood of beech forests in France [1,2]. In decaying wood in Poland, this species is found quite rarely. There are undoubtedly many more such differences in the species diversity of Uropodina inhabiting dead wood in different regions of Europe, which result from different climatic conditions, different geographical ranges, biology, and ecology of individual species, which obviously requires further research.

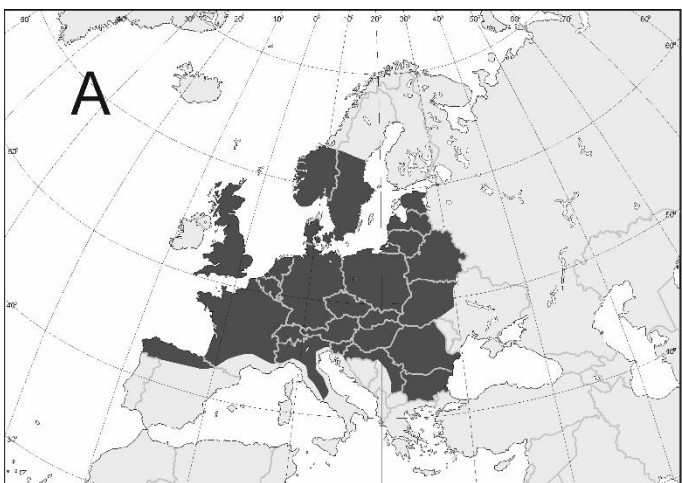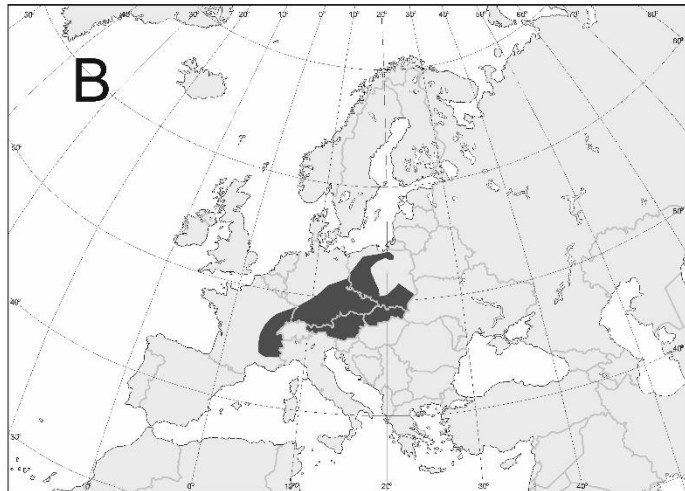

**Figure 4.** Variation of geographic distribution of selected Uropodina species found in dead wood in Europe: (**A**)—*Oodinychus ovalis* (wide distribution), (**B**)—*Olodiscus misella* (narrow distribution) [9].

## 5. Conclusions

As previous studies have shown, the presence of dead wood increases the overall biodiversity of Uropodina in the examined forest ecosystems by about 40% [7]. This microhabitat is therefore very important for the preservation of Uropodina biodiversity. The results presented here show that decaying wood examined in selected provinces of Poland and several other European countries have varied species diversity of Uropodina.

The frequency of these mites in the analyzed dead wood material is low for most species. One of the major factors responsible for the low frequency of mites in analyzed samples, both from Poland and other countries in Europe, is the extent of the research conducted so far, but also the diversity of geographical ranges of individual species [24]. To obtain more reliable results on the species diversity of Uropodina fauna in dead wood microhabitats in Poland and other European countries, further extensive research based on a large number of samples from all of Europe is necessary. This is very important especially in the situation where the phenomenon of a progressive decline in biodiversity has been observed, which also affects mites [33].

**Author Contributions:** Conceptualization, J.B. and A.N.; Methodology, A.N. and J.B.; Software, J.B.; Validation, J.B. and A.N.; Formal Analysis, J.B., A.N. and M.Z.; Investigation, J.B., A.N., M.Z. and M.K.; Resources, J.B., A.N. and M.K.; Data Curation, J.B. and M.Z.; Writing—Original Draft Preparation, J.B., A.N., M.Z. and M.K.; Writing—Review & Editing, A.N., M.K. and M.Z.; Visualization, J.B., M.K. and M.Z.; Supervision, A.N.; Project Administration, J.B. and A.N.; Funding Acquisition, J.B., M.K. and M.Z. All authors have read and agreed to the published version of the manuscript.

**Funding:** This work was possible due to financial support from the Department of General Zoology received for a Ph.D. project carried out by Michał Zacharyasiewicz, M.A and grant No. 049/34/ID-UB/0063.

**Institutional Review Board Statement:** Not applicable.

**Data Availability Statement:** The data presented in this study are stored in an Invertebrate Fauna Bank (Natural History Collections, Faculty of Biology, Adam Mickiewicz University, Poznań, Poland): https://www.wkn.com.pl/wp-content/uploads/2023/03/Katalog-prob%E2%80%93cz.1.pdf (accessed on 20 March 2023).

**Acknowledgments:** The authors of this study are grateful to all who collected dead wood samples which are now stored in the AMU Nature Collections of the Faculty of Biology in Poznań, especially Monika Markowicz, Bartosz Labijak, and Tomasz Rutkowski. The authors of this article are grateful to Natalia Stępczak for proofreading the final manuscript.

**Conflicts of Interest:** The authors declare no conflict of interest.

## Appendix A

*The List of Samples Collected from Deadwood at the Examined Sites in Different European Countries (without Poland)*

France

1–4. Surrounding areas of Bény (46.32050 N 5.269170 E). 27.09.2019. Forest with riparian characteristics, adjacent to a motorway car park. Rotten oak stump. Considerable decay, damp rotten wood. 210 m asl. Leg. J. Błoszyk.

5–8. Argelés-sur-Mer. Camping Taxo les Pins (42.571978 N 3.012747 E). 24.09.2019. Forest with riparian characteristics. Rotten wood from a moderately decayed stump. Leg. J. Błoszyk.

9–12. Argelés-sur-Mer. Camping Taxo les Pins (42.571978 N 3.012747 E). 24.09.2019. Forest with riparian characteristics. Rotten wood from a moderately decayed stump of a poplar. Leg. J. Błoszyk.

13–16. Surrounding areas of Mercurol-Vannes (45.0774 N 4.8888 E). 27.09.2019. Forest with riparian characteristics. Dead wood. Leg. J. Błoszyk.

17–19. Sérignan Plage. Reserve Naturelle des Orpellières (43.2572 N 3.3141 E). 26.09.2019. Dead wood, dry and moderately rotten. Leg. J. Błoszyk.

20–25. Valras-Plage (43.2605 N 3.2554 E). 27.09.2019. Dead poplar. Leg. J. Błoszyk.

Italy

1–10. San Vincenzo. Park (43.081856 N 10.538862 E). 26.03.2019. From lying rotten wood of a cork oak. The wood was well-decomposed and damp. Leg. J. Nowak.

11–20. Bagni San Filippo, Siena province (42.9292 N 11.7034 E). 29.04.2019. From lying rotten wood of a pine. Leg. J. Błoszyk.

Belgium

1. Poppel. Forest Molenheide (51.08035 N 5.39821 E). 28.11.2013. Damp rotten wood. Leg. M. Winkler.

2–7. Brussels. Forest Park (50.8226 N 4.3369 E). Dead wood from a pine. Leg. J. Błoszyk.

8–13. Brussels. Forest Park (50.8226 N 4.3369 E). Dead wood from an oak. Leg. J. Błoszyk.

14–18. Brussels. Duden Park (50.8157 N 4.3318 E). Dead wood from a pine. Leg J. Błoszyk.

19–21. Brussels. Duden Park (50.8157 N 4.3318 E). Dead wood from an oak. Leg J. Błoszyk.

22–23. Brussels. Duden Park (50.8157 N 4.3318 E). Dead wood from a linden. Leg J. Błoszyk.

The Netherlands

1. Kaatsheuvel/ Loon op Zand (51.6431 N 5.05833 E). 29.05.2013. Deciduous forest. Rotten dead wood. Leg. M. Winkler.

2. Tilburg (51.58 N 4.99639 E). 10.05.2013. Forest in the outskirts of the city. Dead pine wood. Leg. M. Winkler.

3. 't Zand (51.498056 N 4.95195 E). 10.08.2013. Mixed forest. Rotten wood of a dead oak. Leg. M. Winkler.

4–7. Loon op Zand (51.6344 N 5.0758 E). 10.10.2013. Deciduous forest. Dead wood. Leg. M. Winkler.

8–11. Loon op Zand (51.6344 N 5.0758 E). 10.10.2013. Deciduous forest. Dead wood of an oak. Leg. M. Winkler.

12–16. Loon op Zand (51.6344 N 5.0758 E). 10.10.2013. Deciduous forest. Damp dead wood of a beech. Leg. M. Winkler.

17–20. Loon op Zand (51.6344 N 5.0758 E). 10.10.2013. Deciduous forest. Rotten stump of a deciduous tree, with traits of considerable wood decay. Leg. M. Winkler.

Slovakia

1. Červený Kláštor (49.3960 N 20.3960 E). 14.04.1987. Spruce forest. Rotten wood from a dead spruce. Leg. L. Miko.

2. Huty (49.2183 N 19.5683 E). 21.08.1988. Fir and beech forest with spruce admixture. Dead wood. Leg. L. Miko.

3–9. Nižný Klátov (48.7335 N 21.1622 E). 16.05.1986. Spruce forest. Dead wood. Leg. J. Błoszyk.

10–15. Dargov (48.7246 N 21.5764 E). 19.05.1986. Spruce forest. Dead wood. Leg. J. Błoszyk.

16–23. Spruce forest North from Bratislava (48.1965 N 17.0964 E). 21.04. 1987. Dead wood. Leg. J. Błoszyk.

24–27. Brezno (48.8700 N 19.6218 E). 16.08.2003. Spruce forest. Dead wood. Leg. J. Nowak.

Denmark

1–8. Copenhagen. Dyrehaven (55.8036 N 12.5664 E). 27.10.2013. Park. Rotten wood from very old oaks. Leg. J. Błoszyk.

8–15. Copenhagen. Dyrehaven (55.8036 N 12.5664 E). 27.10.2013. Park. Rotten wood from very old beeches. Leg. J. Błoszyk.

16–20. Aarhus (56.1846 N 10.1816 E). 10.05.2011. Park. Rotten wood from old oaks. Leg. J. Błoszyk.

21–22. Aarhus (56.1846 N 10.1816 E). 10.05.2011. Park. Rotten wood from old beeches. Leg. J. Błoszyk.

23–26. Møns Klint (54.9642 N 12.5505 E). 13.08.2013. Beech forest. Dead wood. Leg J. Błoszyk.

Sweden

1–6. Ystad (55.5351 N 13.6916 E). 16.07.1989. Mixed forest. Dead wood. Leg. J. Kowalski.

7–11. Malmö (55.5487 N 13.2365 E). 12.06.1990. Deciduous forest. Dead wood. Leg. J. Nowak.

12–18. Lund (55.6848 N 13.2365 E). 9.09.1996. Spruce forest. Dead wood. Leg. Z. Olszanowski.

19–21. Göteborg (57.6924 N 12.0955 E). 11.09.1996. Deciduous forest. Rotten wood from a beech. Leg. Z. Olszanowski.

22–23. Restad (58.3239 N 12.4496 E). 16.08.2003. Spruce forest. Dead wood. Leg. J. Nowak.

Norway

1. Kaupanger (61.1859 N 07.2440 E). 3.06.1998. Deciduous forest, well-developed, species diverse rotten wood of stumps. Leg. M. Gulvik.

2–10. Sogndal, hillside at the fjord, near electric traction (61.2218 N 07.0988 E). 8.07.1999. Spruce, rotten wood from a lying trunk of a spruce, with traits of considerable decay, damp. Leg. J. Błoszyk.

**Table A1.** Percentage participation of dead wood samples from particular tree species in the provinces: Z—Zachodniopomorskie, W—Wielkopolskie, K-P—Kujawsko-Pomorskie, P—Podlaskie, D—Dolnośląskie, Po—Podkarpackie, M—Małopolskie.

| Species | W | D | K-P | M | Po | P | Z |
|---|---|---|---|---|---|---|---|
| *Quercus* L. (oak) | 33.26 | 7.44 | 3.30 | 3.57 | 3.13 | 56.25 | 10.81 |
| *Fagus* L. (beech) | 3.97 | 9.09 | 5.49 | 25.00 | 28.13 | 6.25 | 43.24 |
| *Aesculus* L. (horse-chestnut) | 1.36 | 2.48 | 1.10 | 5.36 | 6.25 | 3.13 | 2.70 |
| *Pinus* L. (pine) | 22.36 | 0.83 | 15.38 | 12.50 | | 6.25 | 27.03 |
| *Tilia* L. (linden) | 2.27 | 4.13 | 4.40 | 1.79 | 21.88 | | 2.70 |
| *Picea* A. Dietrich (spruce) | 0.11 | 45.45 | | 21.43 | 6.25 | 15.63 | 5.41 |
| *Betula* L. (birch) | 6.47 | 6.61 | 4.40 | 1.79 | | 3.13 | |
| *Acer* L. (maple) | 1.25 | 7.44 | 4.40 | 3.57 | | | 2.70 |
| *Acer pseudoplatanus* L. (sycamore) | 0.57 | 0.83 | | 1.79 | 9.375 | | 2.70 |
| *Alnus glutinosa* (L.) Gaertn. (black alder) | 5.90 | 0.83 | 6.59 | | | 6.25 | |
| *Carpinus* L. (hornbeam) | 9.19 | | 3.30 | | 3.13 | 3.125 | |
| *Salix* L. (willow) | 1.36 | 0.83 | | 8.93 | 6.25 | | |
| *Fraxinus* L. (ash) | 0.68 | 0.83 | 1.10 | 1.79 | | | |
| *Robinia pseudoacacia* L. (black locust) | 0.34 | 6.61 | | 1.79 | 3.13 | | |
| *Ulmus* L. (elm) | 0.57 | 5.79 | | | | | |
| *Larix* Mill. (larch) | 9.08 | | | | | | |
| *Sorbus* L. (rowan) | 0.11 | 0.83 | | | | | |
| *Populus* L. (poplar) | 0.91 | | 2.20 | | | | |
| *Populus tremula* L. (aspen) | 2.16 | | | | | | 2.70 |
| *Pyrus* L. (pear tree) | 0.11 | | | | 3.13 | | |
| *Abies* Mill. (fir) | | | | 7.14 | 9.38 | | |
| *Corylus* L. (hazel) | 2.61 | | | | | | |
| *Crataegus* L. (hawthorn) | 0.45 | | | | | | |
| *Sorbus torminalis* (L.) (checker tree) | 0.45 | | | | | | |
| *Platanus* L. (plane) | 0.23 | | | | | | |
| *Sambucus nigra* L. (European elder) | 0.23 | | | | | | |
| *Pseudotsuga* Carrière (Douglas fir) | 0.11 | | | | | | |
| *Malus* Mill. (apple tree) | 0.11 | | | | | | |
| *Prunus* L. (plum) | 0.11 | | | | | | |
| *Prunus spinosa* L. (blackthorn) | 0.11 | | | | | | |
| *Taxus* L. (yew) | | | | 48.35 | | | |
| *Prunus avium* L. (wild cherry) | | | | 1.79 | | | |
| *Cerasus* Mill. (sour cherry) | | | | 1.79 | | | |
| Number of Species | 29 | 15 | 12 | 15 | 11 | 8 | 9 |

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
