# Peer review of "Geographical Differentiation of Mites from the Suborder Uropodina (Acari: Mesostigmata) in Dead Wood in Europe in the Light of Recent Research"

_diversity, doi:10.3390/d15050646_

Round 1

Reviewer 1 Report

This paper provides a summary of a vast number of samples from rotting logs, focusing on the communities of uropodine mites. Although no ecological factors are linked with the patterns observed, the record of diversity and identification of dominant species is useful.  Most species are identified to species, a remarkable testament to the systematic work done on Uropodina in Poland and Europe, most notably from the senior author. 

My main gripe is with the writing of the discussion, which is often vague and overly wordy. I think it could be improved substantially with some care and effort to write clearly and concisely. I have tried to provide advice.

My other criticisms relate to jargon - you will see that I am no fan of "merocenose", which seems like jargon of the worst kind (i.e. you could remove it and the paper would be easier to understand). One paragraph of the Results also seemed unnecessary as the figures already presented the data clearly.

These faults can be overcome. This is quality work, reporting on an important but understudied habitat of ecological and evolutionary significance (i.e. rotting logs). The work is substantial and an excellent reference point for future studies in Europe, and may become interesting to compare with international studies. I hope to see it published soon.

Author Response

Responses to Reviewer 1

Geographical differentiation of mites from the suborder Uropodina (Acari: Mesostigmata) in dead wood in Europe and Poland in the light of recent research

The authors of the study are grateful to the Reviewer for all comments and suggestions. All of them have turned out to be extremely helpful, which obviously has considerably improved the overall quality of the manuscript.

Detailed responses to the Reviewer comments:

This paper provides a summary of a vast number of samples from rotting logs, focusing on the communities of uropodine mites. Although no ecological factors are linked with the patterns observed, the record of diversity and identification of dominant species is useful.  Most species are identified to species, a remarkable testament to the systematic work done on Uropodina in Poland and Europe, most notably from the senior author. 

My main gripe is with the writing of the discussion, which is often vague and overly wordy. I think it could be improved substantially with some care and effort to write clearly and concisely. I have tried to provide advice.

  • The remarks and suggestions concerning Discussion have been taken into consideration

My other criticisms relate to jargon - you will see that I am no fan of "merocenose", which seems like jargon of the worst kind (i.e. you could remove it and the paper would be easier to understand). One paragraph of the Results also seemed unnecessary as the figures already presented the data clearly.

  • The term 'merocenose' has long been used in the literature on dead wood and other microhabitats. However, as suggested by the reviewer, it has been removed from the text.

  • The commentary on the similarities of the communities in the provinces has been retained, as we believe that the figures should be commented on. On the other hand, we have added a list of the trees from which the dead wood came in each province (Table A1). The habitat type, i.e. the tree stand, is not so very important for the species composition of Uropodina in the dead wood, as the rotten wood was not always collected from the dominant species in the tree stand, what is more important is the tree species from which it came (Zacharyasiewicz et al. 2021, DOI: 10.3390/d13120609).

These faults can be overcome. This is quality work, reporting on an important but understudied habitat of ecological and evolutionary significance (i.e. rotting logs). The work is substantial and an excellent reference point for future studies in Europe, and may become interesting to compare with international studies. I hope to see it published soon.

Other Reviewer’s comments from the Text:

Also: I would argue that dead wood is not an unstable microhabitat, in the sense that it lasts an extraordinary amount of time compared to others patchy habitats (dung, carrion, puddles, for example). How long does it take a log to decay in a boreal forest? 20+ years?

-         The term "unstable microhabitat" has long been used in the acarological literature to refer to dead wood, but also to micro-environments such as bird nests, mammal nests, anthills, animal droppings and others. These microhabitats are "unstable" in comparison to the soil and, although they can function for a long time, their impermanence is mainly expressed in the fact that the wood is transformed (decomposition process continues) and at each stage it is populated by different fauna.

Was there any effort to standardise the amount of material put into Tullgren funnels? Or the heat of the bulbs? (I think, for this study, it's ok if it wasn't - but if you did use a standard procedure then it should be explained so that others can do exactly the same thing)

  • The samples from the dead wood were always similar in volume, ranging from 0.5-0.8 litres, and were always extracted for a similar period of time, depending on the moisture level of the material 4-6 days). This information has been added in the chapter "Data collection". We also use the same voltage bulbs constantly in the tullgrens.

Reviewer 2 Report

Authors of that manuscript investigated species composition and diversity of mites (Acari) from suborder Uropodina inhabiting dead wood. To investigate geographical variation in Uropodina distribution and diversity the authors analyzed data from samples they collected in seven regions in Poland and in eight other European countries. The concept of such work I consider to be highly original and very interesting. I would also like to add that the manuscript title is very catchy and the abstract is well written, which surely will attract attention from readers interested in biodiversity and biogeography. Unfortunately, to my high disappointment, beyond that I am not really able to complement that work. Bellow I present my biggest concerns about that work with some suggestions for its improvements.

Introduction

As the authors have stated "dead wood is an important element in increasing the biodiversity of forest ecosystems, both for the mites in question and for other groups of organisms [3,4,13,14,15,16,17,18,19,20,21,22,23]." I would even say that dead wood provides essential habitat for a very high diversity of various biota and is essential driver of ecosystem processes. There is enormous amount of literature devoted to these topics. Unfortunately, authors has quoted only one general literature position on that and all the remaining is virtually limited to their own literature on mites or some insignificant publications (in Polish) or a conference presentation, thus virtually completely ignoring vast knowledge on significance of dead wood for forest ecosystems and its diverse biota. In that context, it is also a bit surprising that over 1/3 of cited references (i.e., 12 out of 32) are previous works of the authors. I do understand that the authors are specialists on Uropodina and maybe even most publications on that group were published by them, but ignoring general context of their work seriously limits value of the manuscript. I would therefore, suggest to elaborate these topics and provide a separate paragraph devoted to that.

Material and Methods

2.1. Study area.

While the work lists provinces in Poland and other European countries where samples were collected it lacks essential information on exact location of the sampling sites. Without such information it is impossible to evaluate if the sampling provided any representative data therefore it is important to provide at least a map showing exact location of the sampling sites or preferably a table with GPS coordinates and number of samples collected from particular location and substratum type. From previous literature of authors it seems to me that in some regions all the samples were probably collected from relatively small area, so they might be not really representative for a particular province or country. Moreover, information on sample size is also essential. Without that data given in Tables 2-8, particularly “Maximum number of species per sample" are completely meaningless and if there are very big differences in size of particular samples might be misleading. Hopefully authors did not omit such information on purpose to hide week points of their sapling design. Anyway, without such information it is not possible to evaluate if the work was done properly and if results are truly meaningful.

2.2. Data collection.

Title of that section is misleading. This section does not provide any information on data collection but only on sample treatment and processing.

Material and Methods

Line 117: There is inconsistency in between information given in the text and Table 1. The table lists 17 species whereas in the test it is written that 16 species were found in Zachodnipomorskie.

Table 2-8.

The Table captions lists abbreviation "Awer" whereas the Tables’ first row "Aver".

Some species are described as sp. n., e.g. C. cassiseasimilis sp. n. or C. rafalski sp. n., as these species are not described for the first time in that work adding the abbreviation "sp. n." is inappropriate.

Table 3.

It seems that the second column is two narrow and some numbers given it that column are split into two or even three rows.

Line 222: "Samples of dead wood collected in several European countries are stored in the … collection"

As I understand, and as it is written in methods, the collected samples of dead wood were processed to extract mites and the extracted mites are stored in the collection (not dead wood samples).

Figure 3.

What is the source of data presented on that Figure? Are these distribution ranges based on the authors’ collection or literature sources, or both? There is nothing on that in Material and Methods section. If fact, there is even nothing on that in Results section and Figure 3 is only cited in Discussion. Source of these data have to be clarified.

Lines 249-250:"… the number of species was equal (ranging from between 23 and 27)."

No, the number of 23 is not equal to 27. This has to be corrected.

Discussion

Line 283: "Be it noted … " -> "It is worth (important) to note … " reads better.

Information given in that paragraph (based on authors´ previous publications) emphasizes the need for detailed information on sampling design. Since, "frequency of mires … are (is) highly dependent on tree species …" (see line 286) as well as "the degree of wood decay and its moisture content" (see line 289) data without such information have very limited value and cannot be interpreted properly.

Lines 301-303: "The presented studies show that such widely distributed species as D. carinatus, P. pulchella, U. pyriformis, O. minima, O. ovalis, T. aegrota and Trematurella elegans."

This sentence is clearly unfinished or grammatically incorrect.

Lines 308-309: "such species … have their northern limit in Poland."

Source of that information is needed.

Lines 310-311:

"The group of species with different ranges of occurrence also includes Polyaspinus schweizeri, which is an east-Carpathian species … "

Source of that information is needed.

Conclusions

Line 344: "The results presented here to show …"

Grammar correction needed.

Data Availability Statement:

Lines 365-366: "The data presented in this study stored in a computer database called AMUNATCOLL and openly available at: https://amunatcoll.pl/ (accessed on 15 February 2023)."

This information is not correct and misleading. This database contains only species records, so it does not store complete data of that work and does not allow for verification of presented results. This statement should precisely define what is stored in the database; otherwise the statement is simply false and misleading.

References

As I have already indicated over 1/3 of cited references (i.e., 12 out of 32) are previous works of the authors. I do understand that the authors are specialists on the investigated biota and maybe even most publications on that group were published by them, still such high proportion of authors own publication among cited references "raises a red flag" suggesting inappropriate or even unethical practices. Moreover, ignoring general context of the work demonstrates that authors did not seriously consider their work.

As regards terminology, I do not thing that using "community" for a regional faunas is appropriate. The term "community" always implies some kind of interactions among biota forming the community. In this case there not only no evidence on any kind of interactions, but since many of the listed species inhabit different microhabitats, it is virtually certain that there is no interactions among these species at all. Thus the "fauna" should be used insted.

I would also suggest unifying the form of names for "Malopolska/Małopolskie" and "Wielkopolska/Wielkopolskie" regions. At present these two different forms are used interchangeable. Considering the form of the other Polish regions I would suggest using accordingly "Małopolskie" and "Wielkopolskie".

Moreover, it seems to me some phrases and language style is sometimes a kind of awkward, thus grammar and language need detail verification and correction. However, as English is not my mothers’ tongue I do not feel confident providing details comments and correction. This is not reviewers’ role anyway.

Considering all the above, particularly the lack of essential information on methodology, I was wondering whether I should recommend rejection of the manuscript. However, considering originality and novelty of this work, and its potentially high interests I would suggest reconsider its publication after major revision with detailed consideration and addressing all the issues listed above.

Author Response

Responses to Reviewer 2

Geographical differentiation of mites from the suborder Uropodina (Acari: Mesostigmata) in dead wood in Europe and Poland in the light of recent research

The authors of the study are grateful to the Reviewer for all comments and suggestions. All of them have turned out to be extremely helpful, which obviously has considerably improved the overall quality of the manuscript.

Detailed responses to the Reviewer comments:

Introduction

As the authors have stated "dead wood is an important element in increasing the biodiversity of forest ecosystems, both for the mites in question and for other groups of organisms [3,4,13,14,15,16,17,18,19,20,21,22,23]." I would even say that dead wood provides essential habitat for a very high diversity of various biota and is essential driver of ecosystem processes. There is enormous amount of literature devoted to these topics. Unfortunately, authors has quoted only one general literature position on that and all the remaining is virtually limited to their own literature on mites or some insignificant publications (in Polish) or a conference presentation, thus virtually completely ignoring vast knowledge on significance of dead wood for forest ecosystems and its diverse biota. In that context, it is also a bit surprising that over 1/3 of cited references (i.e., 12 out of 32) are previous works of the authors. I do understand that the authors are specialists on Uropodina and maybe even most publications on that group were published by them, but ignoring general context of their work seriously limits value of the manuscript. I would therefore, suggest to elaborate these topics and provide a separate paragraph devoted to that.

-         As the reviewer rightly pointed out, the authors have already devoted a considerable number part of their previous works to assess the role of dead wood in shaping biodiversity, especially in relation to Uropodina communities in forest ecosystems. The work presented here addresses aspects of this problem that have not yet been discussed. We considered it unnecessary to reproduce here the generally known information on the role of dead wood. Not all of the publications cited in this section are papers by the authors; the others deal with different groups of mites and other invertebrates. The fact that some have been published in Polish does not mean that they are less important, as they contribute important information on the whole discussion. However, as suggested, we have also added more publications discussing the importance of dead wood for other organisms.

Material and Methods

2.1. Study area.

While the work lists provinces in Poland and other European countries where samples were collected it lacks essential information on exact location of the sampling sites. Without such information it is impossible to evaluate if the sampling provided any representative data therefore it is important to provide at least a map showing exact location of the sampling sites or preferably a table with GPS coordinates and number of samples collected from particular location and substratum type. From previous literature of authors it seems to me that in some regions all the samples were probably collected from relatively small area, so they might be not really representative for a particular province or country. Moreover, information on sample size is also essential. Without that data given in Tables 2-8, particularly “Maximum number of species per sample" are completely meaningless and if there are very big differences in size of particular samples might be misleading. Hopefully authors did not omit such information on purpose to hide week points of their sapling design. Anyway, without such information it is not possible to evaluate if the work was done properly and if results are truly meaningful.

-      In Data Availability Statement we added the right link, which refers to „Katalog prób glebowych i prób z mikroÅ›rodowisk a Zbiorach Przyrodniczych WydziaÅ‚u Biologii UAM”, BÅ‚oszyk et al. 2023. A map of the sites in Poland (Figure 1) and a list of samples from the surveyed European countries have also been added in the Appendix. Furthermore, we do not overlook the fact that the extent of survey in individual provinces in Poland is uneven - the number of collected samples is given and we refer to this in Discussion.

2.2. Data collection.

Title of that section is misleading. This section does not provide any information on data collection but only on sample treatment and processing.

 - The relevant information has been added in this part of the manuscript.

Material and Methods

Line 117: There is inconsistency in between information given in the text and Table 1. The table lists 17 species whereas in the test it is written that 16 species were found in Zachodnipomorskie.

  • The number of species in Table 1 has been corrected.

Table 2-8.

The Table captions lists abbreviation "Awer" whereas the Tables’ first row "Aver".

  • Tables have been corrected

Some species are described as sp. n., e.g. C. cassiseasimilis sp. n. or C. rafalski sp. n., as these species are not described for the first time in that work adding the abbreviation "sp. n." is inappropriate.

  • Corrected

Table 3.

It seems that the second column is two narrow and some numbers given it that column are split into two or even three rows.

- It must have happened during formatting of the table. The comment will be sent to the Editorial office.

Line 222: "Samples of dead wood collected in several European countries are stored in the … collection"

As I understand, and as it is written in methods, the collected samples of dead wood were processed to extract mites and the extracted mites are stored in the collection (not dead wood samples).

  • In the Natural History Collections at AMU, we keep as separate collections both the mites already extracted from the samples and identified, as well as the whole extracted samples preserved in alcohol. In most cases that we are aware of, researchers discard the samples after selecting the soil fauna of interest from them; we retain the collected material to be able to return to it in the event of later studies on other groups. This way of proceeding makes it possible to maximise the use of once-collected material by many researchers. So far, on the basis of the collected sample collection, there have been a number of monographic studies of different mite groups and hundreds of publications describing mite species from different groups that are new to knowledge, published by different authors using the same material. The collection, accumulated for more than 70 years, includes more than 40,000 samples from different environments from all continents.

Figure 3.

What is the source of data presented on that Figure? Are these distribution ranges based on the authors’ collection or literature sources, or both? There is nothing on that in Material and Methods section. If fact, there is even nothing on that in Results section and Figure 3 is only cited in Discussion. Source of these data have to be clarified.

  • The source of the Figure has been added in the caption.

Lines 249-250:"… the number of species was equal (ranging from between 23 and 27)."

No, the number of 23 is not equal to 27. This has to be corrected.

  • Yes, it is not equal but it is similar.

Discussion

Line 283: "Be it noted … " -> "It is worth (important) to note … " reads better.

Information given in that paragraph (based on authors´ previous publications) emphasizes the need for detailed information on sampling design. Since, "frequency of mires … are (is) highly dependent on tree species …" (see line 286) as well as "the degree of wood decay and its moisture content" (see line 289) data without such information have very limited value and cannot be interpreted properly.

  • The paragraph has been rephrased. Information about preferences to tree species and humidity is given.

Lines 301-303: "The presented studies show that such widely distributed species as D. carinatus, P. pulchella, U. pyriformis, O. minima, O. ovalis, T. aegrota and Trematurella elegans."

This sentence is clearly unfinished or grammatically incorrect.

  • The sentence has been corrected.

Lines 308-309: "such species … have their northern limit in Poland."

Source of that information is needed.

  • Source has been added.

Lines 310-311:

"The group of species with different ranges of occurrence also includes Polyaspinus schweizeri, which is an east-Carpathian species … "

Source of that information is needed.

  • Source has been added.

Conclusions

Line 344: "The results presented here to show …"

Grammar correction needed.

  • The sentence has been corrected.

Data Availability Statement:

Lines 365-366: "The data presented in this study stored in a computer database called AMUNATCOLL and openly available at: https://amunatcoll.pl/ (accessed on 15 February 2023)."

This information is not correct and misleading. This database contains only species records, so it does not store complete data of that work and does not allow for verification of presented results. This statement should precisely define what is stored in the database; otherwise the statement is simply false and misleading.

-      In Data Availability Statement has now the correct link referring to „Katalog prób glebowych i prób z mikroÅ›rodowisk a Zbiorach Przyrodniczych WydziaÅ‚u Biologii UAM”, BÅ‚oszyk et al. 2023: https://www.wkn.com.pl/katalog/katalog-prob-glebowych-i-prob-z-mikrosrodowisk-w-zbiorach-przyrodniczych-wydzialu-biologii-uam-czesc-i-proby-z-polski-z-akronimem-pl-zebrane-w-latach-1938-1999-recenzowan/

References

As I have already indicated over 1/3 of cited references (i.e., 12 out of 32) are previous works of the authors. I do understand that the authors are specialists on the investigated biota and maybe even most publications on that group were published by them, still such high proportion of authors own publication among cited references "raises a red flag" suggesting inappropriate or even unethical practices. Moreover, ignoring general context of the work demonstrates that authors did not seriously consider their work.

As regards terminology, I do not thing that using "community" for a regional faunas is appropriate. The term "community" always implies some kind of interactions among biota forming the community. In this case there not only no evidence on any kind of interactions, but since many of the listed species inhabit different microhabitats, it is virtually certain that there is no interactions among these species at all. Thus the "fauna" should be used insted.

I would also suggest unifying the form of names for "Malopolska/Małopolskie" and "Wielkopolska/Wielkopolskie" regions. At present these two different forms are used interchangeable. Considering the form of the other Polish regions I would suggest using accordingly "Małopolskie" and "Wielkopolskie".

  • The names of provinces have been corrected.
  • The large number of articles cited by the authors is due to the fact that there are currently no other studies on Uropodina groupings in dead wood in such detail. It would then be difficult to omit these items on purpose. On the other hand, following the reviewer's suggestion, we have added publications by other authors on the issue of dead wood in Introduction.
  • The term “community” is often used in the acarological literature.

Moreover, it seems to me some phrases and language style is sometimes a kind of awkward, thus grammar and language need detail verification and correction. However, as English is not my mothers’ tongue I do not feel confident providing details comments and correction. This is not reviewers’ role anyway.

  • The manuscript has been proofread by a native speaker and corrected.

Considering all the above, particularly the lack of essential information on methodology, I was wondering whether I should recommend rejection of the manuscript. However, considering originality and novelty of this work, and its potentially high interests I would suggest reconsider its publication after major revision with detailed consideration and addressing all the issues listed above.

Reviewer 3 Report

Manuscript: Geographical differentiation of mites from the suborder Uropodina (Acari: Mesostigmata) in dead wood in Europe and Poland in the light of recent research

Although there are some useful elements in this study, many aspects are problematic and there is a need for an overall careful re-write. The work is largely focused on samples from Poland but very little is said about the rest of the other European countries object of the study.

The Introduction did not provided any useful elements to guide the reader through the subject of the article and just made a list of previous studies of the same authors in the same areas. In addition, the repetitions throughout the text are making the article difficult to read. This section can be easily implemented by exploring, for example, the ecology of the Uropodina.

Although this work includes many samples, Materials and Methods are presented in overall superficial and in some parts incomplete way. This section can also be implemented including more information about the sampling methods and presenting the data in a well organised manner. The same can be said for the Results section: several mistakes still presents (e.g. spelling of the species scientific names, “copy-paste” mistakes in all the Tables descriptions) but everything can be corrected and implemented. The Discussion and Conclusions sections are overall weak in the structure and the claims are often not supported but this can be adjusted in the light of a solid introduction.

General comment: the phrase “Uropodina communities inhabiting dead wood” (and all its variations) has been used in almost 70% of the sentences in this article. I suggest to pay more attention at the repetitions while write the text.

Lines 2-4: Title. I would rephase it “Geographical distribution of mites from the suborder Uropodina (Acari: Mesostigmata) in dead wood in Europe.”

Line 15: Poland is in Europe. Remove Poland.

1. Introduction

Very weak introduction. It can be implemented with more solid examples and arguments. I would include information about the ecology of the Uropodina to better understand the results of the article itself.

The text can be improved also from the sentences constructions point of view. To many repetitions are making the text difficult to read.

Lines 34-36: I agree with the statement but I would support it with the existing literature and add some references.

Lines 36-37, 41: the research…her research…the research. I would rephrase it

Lines 41-43: I would rephrase it as: “For many years, several studies…..” and remove “by BÅ‚oszyk and his co-workers”, the references are enough.

Line 44: If there are any results in those papers relevant to this article, I would name them, otherwise remove "among other things".

Line 44: found…found

Lines 45-46: What are the arguments that led the authors to this conclusion? Is what listed above enough to conclude the importance of the dead wood in forest ecosystems? I would add examples before jumping to this conclusion.

Line 50: published studies but only one study is cited in the references.

Lines 47-50: I would move it before the conclusion of lines 45-46.

Lines 48, 51, 54, 59, 61, 63, 65: communities inhabiting different types dead wood… communities occurring dead wood…communities inhabiting dead wood…communities inhabiting dead wood […]. All these repetitions are making the text a bit tedious to read. I suggest rephrasing the sentences carefully. Use synonyms (e.g. decayed wood…)

2. Materials and methods

2.1. Study area

The locations listed this way have a little meaning. One area can be different from the other, therefore also the biodiversity composition. Describing and listing the habitats where the samples were collected is fundamental also to the interpretation of the results.

The explanation of the polish regions is not needed and could be easily replaced with a map.

A map will also give a better idea of the distribution of the sampling effort.

I also recommend to create a table for the polish and other European sampling sites with the name, ID abbreviations (like the ones present in Figure 2 of the results), the type of habitat and the coordinates.

Lines 70-71: collected…collected.

Line 71: Itay. Correct “Italy”

Line 72: Netherlands. Correct “the Netherlands”.

Lines 73-74: Obviously are in different geographical places...this sentence is redundant and it has to be better rephrase it.

Lines 74-75: How many samples for each province were collected? I would add the samples number as done for the European countries.

Lines 75-80: all this description is unnecessary because a map can easily show it.

2.2. Data collection

The data collection section is incomplete because it refers only to the polish samples. It should be mention in the text where all the other European samples were collected, extracted and preserved if the procedure differs from the polish samples. Only at page 13 it is mentioned that the Samples of dead wood collected in several European countries are stored in the Natural History Collections but without naming the collections.

Line 82: When did you collect the samples? Can communities composition vary throughout the seasons?

Lines 82-83: I would remove the brackets and rephrase it “depending on the humidity in the soil samples”

Line 84: ethyl alcohol. Ethanol

Line 85: Please mind the verb tense consistency.

Lines 87-88: The identification of 87 the species was carried out by the first author. Probably, this can be moved to the “Author Contributions” section.

Lines 88-90: Did you prepared permanent slides with another medium or the specimens were left in the lactic acid?

2.3. Data analysis

Line: 91: Which keys did you use for the species identification?

Lines 93-97: To make it easier to read, I'd transfer these categories into a table or bulleted list.

Lines 98-99: The community similarity in the communities found in dead wood in the examined provinces. I would rephrase it.

Line 102: (PoznaÅ„, Poland) the period is missing at the end of the sentence “(PoznaÅ„, Poland).”

3.Results

3.1. Geographical variation of Uropodina communities in Poland

Lines 105-107: I would rephrase it “52 Uropodina species were found in the examined merocenoses of the seven polish provinces. Only seven (13.5%) species occurred in all the provinces.”

Line 110: I would include in the text of the results also the percentages of the most abundant species across the provinces. It will give a clear picture of the findings at first glance.

Line 112: Table 1 I would list the species in this table grouping them according to the Family. I would also recommend to change the page orientation from vertical to horizontal to make the table easier to read (the species name will be then on one line).

Other suggestion: Figure 2 of the results already has ID abbreviations of the provinces. I would use the same here.

Carefully read all the species and correct the scientific names and authorship!

It is better to use the accepted species name rather than the synonym. Some of the names are misspelled.

Please refer to the International Commission on Zoological Nomenclature.

After correcting this, please carefully check also the other results tables (Table 2 to Table 9)

Below some example:

·       Dinychus carinatus A.Berlese, 1903

·       Olodiscus minima (Kramer, 1882): Uropoda minima Kramer, 1882(accepted name),Olodiscus minimus Kramer, 1882 (synonym)

·       Trachytes aegrota (Koch, 1841)

·       Trematurella elegans (Kramer, 1882): is this the correct attribution?

·       Dinychus woelkei W.Hirschmann Zirngiebl-Nicol, 1969

·       Trachytes pauperior (Berlese, 1914)

·       Urodiaspis tecta (P.Kramer, 1876)

·       Dinychus arcuatus (Trägårdh, 1943)

·       Discourella (?) baloghi: why there is a question mark?

·       Polyaspis snasonei Berlese, 1916. Polyaspis sansonei Berlese, 1916

·       …and so on…

Lines 113-114: I would include for each following provinces paragraphs, the specification of the habitat were the species occurred.

Line 118: (Table 2 ). Remove space after 2

Lines 123, 135, 144, 153, 161, 170, 178Awer±SD. Correct the spelling. Be consistent in the table: if in the table caption are together, then in the table should be only one column and the SD should have the ± in front of the values.

Line 135: Wielkopolskie or Wielkopolska? Be consistent all along the text.

N – number of specimens. In the table if the number is e.g. 9421 it should go all in the same line. Expand the size of the column.

Line 158: remove double space.

Line 170: MaÅ‚opolskie or MaÅ‚opolska? Be consistent all along the text.

3.2. Geographical similarity of species composition of examined communities

Lines 188-191: communities inhabiting dead wood. Avoid to many repetitions

Lines 196-212: All the names are making the text hard to read and there is no correspondence with the codes showed by Figure 2. After naming the provinces the first time in the paragraph of the Study area, and introducing the ID codes, I suggest to use the same codes also in the text.

Line 214: Figure 1 I would use ID codes of the provinces to make the tree more compact and easier to read. Then, a legenda with the code names can be added in the figure description.

Line 218: What kind of software was used to prepare these maps? Included it in the figure description

3.3. Species diversity of Uropodina communities in dead wood in some European countries

Lines 222-223: which collections? This should be mention in the Materials and methods paragraph.

Lines 225-226: Was the sampling method and the extraction across all the samples comparable? Is it possible that the low number of specimens in the wood samples is due to a different extraction procedure or a different preservation of the samples itself? Was the sampling method and the extraction similar to the polish samples?

Line 228: the Netherlands

Line 229: only Norwegian samples are based on materials collected by the authors? This should be also specified in the Materials and methods.

Line 240: In Figure 3 A there is a yellow island (?) to the west of the Portugal coast. Please correct the figure.

I would enlarge the focus on the countries object of the study and remove all the meridians and parallels so that the picture will be more clear. The degrees on the frame are self-explanatory.

What kind of software was used to prepare these maps? Included it in the figure description

Lines 243-244: Why unconfirmed occurrence is reported on the map with specimens found in the European dead wood merocenoses? I think this map is misplaced: it should not be at the end of the Results but as part of the Discussion. I would also make more evident the colour of the assessed geographical ranges in contrast with the unconfirmed occurrence (e.g. black/light grey, blu/yellow; red/yellow…).

4.Discussion

Line 245: insert space between 4. and Discussion

Line 257: aforementioned factors. Which factors? Nothing was previously mentioned.

Lines 257-258: Without knowing the exact sampling procedure, it is difficult to assume anything. It is fundamental for the paper to be thorough and provide details of the data collection. Only then, it will be possible to discuss whether or not the modeling of the research is responsible for these findings.

Lines 259-260: Quantity is not always the answer. A solid and standardized sampling system can also help in the data interpretation.

Lines 262-263: not only due to their species composition, but also their community structure. Are you talking about the species composition and community structure of the Uropodina or of the unstable microhabitats?

Lines 264-265: was it ever consider that maybe this is connected to the food resources presents in different habitats? If so, I would cite some examples.

Lines 267-268: whole community [8]. A similar structure can be observed in the analysed 267 communities inhabiting dead wood from the seven surveyed provinces in Poland. The cited work (NapieraÅ‚a and BÅ‚oszyk, 2013) is also based on polish samples…so can you say that the structure is similar to itself? I would find another paper that covers other regions or I would think a better way to phrase these two sentences.

Lines 269-272: the most dominant species were O. ovalis and P. pulchella but they were only dominant in 3 out of 7 provinces investigated? To better highlight the findings, I would rephrase the sentences not form provinces perspective but from the species perspective.

E.g. O. ovalis was one of the most dominant species in all seven provinces (from 19,15% up to 63,12%).

Reading the results, P. pulchella was one of the dominant species only in 3 provinces out of 7.

Line 274-278: more samples more species, less samples less species. How can you state that a low frequency had impact on the species diversity?? undoubtedly had an impact. What impact? In which way had an impact? Isn’t maybe a problem with the sampling?

This part of the discussion does not make sense.

Line 280-281, 283: species…species…species…species. Rephrase it.

Lines 283-284: of Uropodina communities inhabiting dead wood. I would remove it. This sentence is extremely frequent and makes the text tedious to read.

Line 284: other conditions found previously. What conditions? What are you referring to?

Lines 283-298: all this part should be in the Introduction! Nothing in the introduction was told about the ecology of the Uropodina! It is a fundamental part to interpretate the data correctly and be able to discuss it!

Lines 302-305: some species aren’t abbreviated. Why? Were all the species already mentioned at least one time in the previous text?

Lines 306-308: ?

Lines 311-316: some species aren’t abbreviated. Why? Were all the species already mentioned at least one time in the previous text?

Lines 319-339: through the entire paper no little relevance was given to European samples. If the results aren’t relevant and the samples not sufficient…why are included in the article. I would consider, at this point, to remove from the analysis the European samples all together or put a lot more effort in the description, the methods, the results and the discussion.

5. Conclusions

Based on the introduction and the discussion, the conclusions are also a bit weak.

Lines 344: The results presented here to show. Remove to.

Line 355: Author Contributions Please check carefully again. There are many mistakes in the punctuation.

Line 371: References Please check carefully the correspondence of the cited articles with the text and be consistent in the references format.

Author Response

Responses to Reviewer 3

Geographical differentiation of mites from the suborder Uropodina (Acari: Mesostigmata) in dead wood in Europe and Poland in the light of recent research

The authors of the study are grateful to the Reviewer for all comments and suggestions. All of them have turned out to be extremely helpful, which obviously has considerably improved the overall quality of the manuscript.

Detailed responses to the Reviewer comments:

General comment: the phrase “Uropodina communities inhabiting dead wood” (and all its variations) has been used in almost 70% of the sentences in this article. I suggest to pay more attention at the repetitions while write the text.

Lines 2-4: Title. I would rephase it “Geographical distribution of mites from the suborder Uropodina (Acari: Mesostigmata) in dead wood in Europe.”

Line 15: Poland is in Europe. Remove Poland.

  • Title has been changed.

  1. Introduction

Very weak introduction. It can be implemented with more solid examples and arguments. I would include information about the ecology of the Uropodina to better understand the results of the article itself.

The text can be improved also from the sentences constructions point of view. To many repetitions are making the text difficult to read.

Lines 34-36: I agree with the statement but I would support it with the existing literature and add some references.

Lines 36-37, 41: the research…her research…the research. I would rephrase it                     

Lines 41-43: I would rephrase it as: “For many years, several studies…..” and remove “by BÅ‚oszyk and his co-workers”, the references are enough.

Line 44: If there are any results in those papers relevant to this article, I would name them, otherwise remove "among other things".

Line 44: found…found

Lines 45-46: What are the arguments that led the authors to this conclusion? Is what listed above enough to conclude the importance of the dead wood in forest ecosystems? I would add examples before jumping to this conclusion.

  • Other citations on the role of dead wood in the forest ecosystem have been added in the following sentence.

Line 50: published studies but only one study is cited in the references.

Lines 47-50: I would move it before the conclusion of lines 45-46.

Lines 48, 51, 54, 59, 61, 63, 65: communities inhabiting different types dead wood… communities occurring dead wood…communities inhabiting dead wood…communities inhabiting dead wood […]. All these repetitions are making the text a bit tedious to read. I suggest rephrasing the sentences carefully. Use synonyms (e.g. decayed wood…)

  • All the corrections suggested by the reviewer in Introduction have been made. However, it is difficult to avoid repetition when the article is about dead wood. We have tried as much as possible to replace the term with synonyms.

  1. Materials and methods

2.1. Study area

The locations listed this way have a little meaning. One area can be different from the other, therefore also the biodiversity composition. Describing and listing the habitats where the samples were collected is fundamental also to the interpretation of the results.

  • The aim of this study is not to analyse Uropodina communities in terms of ecological conditions, but to show the geographical variation and the current status of the research in different regions of Poland and European countries. Ecological issues such as the dependence of the composition of the communities on the species of dead wood (Zacharyasiewicz et al. 2021), the type of merocenosis or other factors i.e. temperature, humidity and degree of wood decomposition (BÅ‚oszyk et al. 2015) have been analysed in previous publications.

The explanation of the polish regions is not needed and could be easily replaced with a map.

A map will also give a better idea of the distribution of the sampling effort.

  • The relevant map has been added.

I also recommend to create a table for the polish and other European sampling sites with the name, ID abbreviations (like the ones present in Figure 2 of the results), the type of habitat and the coordinates.

Both in the manuscript and in „Data Availability Statement” we added the right link referring Katalog prób glebowych, in which the samples from Poland used for these analyses are listed. Samples from other countries are listed in one of the tables.

Lines 70-71: collected…collected.

Line 71: Itay. Correct “Italy”

Line 72: Netherlands. Correct “the Netherlands”.

  • Linguistic correction have been made.

Lines 73-74: Obviously are in different geographical places...this sentence is redundant and it has to be better rephrase it.

  • We have left this sentence as it is to emphasise that the samples came from geographically diverse regions of Poland, which relates to the main aim of the work.

Lines 74-75: How many samples for each province were collected? I would add the samples number as done for the European countries.

  • The samples number have been added in brackets.

Lines 75-80: all this description is unnecessary because a map can easily show it.

  • The descriptions have been removed.

2.2. Data collection

The data collection section is incomplete because it refers only to the polish samples. It should be mention in the text where all the other European samples were collected, extracted and preserved if the procedure differs from the polish samples. Only at page 13 it is mentioned that the Samples of dead wood collected in several European countries are stored in the Natural History Collections but without naming the collections.

  • All samples used in the study are now stored in Natural History Collections of the Faculty of Biology in PoznaÅ„, which contains the world's largest collection of soil samples. Detailed information on the collection can be found in the Soil Samples Catalogue, available at https://www.wkn.com.pl/katalog/katalog-prob-glebowych-i-prob-z-mikrosrodowisk-w-zbiorach-przyrodniczych-wydzialu-biologii-uam-czesc-i-proby-z-polski-z-akronimem-pl-zebrane-w-latach-1938-1999-recenzowan/

Line 82: When did you collect the samples? Can communities composition vary throughout the seasons?

  • The number of the samples used in the analyses is very high and they cover all seasons. In addition, in the case of communities from different trees, the season of sample collection is not relevant.

Lines 82-83: I would remove the brackets and rephrase it “depending on the humidity in the soil samples”

  • Brackets have been removed.

Line 84: ethyl alcohol. Ethanol

Corrected.

Line 85: Please mind the verb tense consistency.

  • The sentence has been corrected.

Lines 87-88: The identification of 87 the species was carried out by the first author. Probably, this can be moved to the “Author Contributions” section.

  • We always provide this type of information here, as it is not possible to include this information in the “Author Contributions” section.

Lines 88-90: Did you prepared permanent slides with another medium or the specimens were left in the lactic acid?

  • Temporary slides were only made out of some difficult to identify rare species or juvenile stages. After the designation, the mites were transferred to alcohol and are still stored in this form. More information at: https://www.wkn.com.pl/katalog/katalog-prob-glebowych-i-prob-z-mikrosrodowisk-w-zbiorach-przyrodniczych-wydzialu-biologii-uam-czesc-i-proby-z-polski-z-akronimem-pl-zebrane-w-latach-1938-1999-recenzowan/

2.3. Data analysis

Line: 91: Which keys did you use for the species identification?

  • These information has been given Data collection section (line 91).

Lines 93-97: To make it easier to read, I'd transfer these categories into a table or bulleted list.

  • These categories have been used several times by us in publications in MDPI in the body of the text, so we decided to leave as it is (for example: https://doi.org/10.3390/f13081219, https://www.mdpi.com/1424-2818/13/10/476 )

Lines 98-99: The community similarity in the communities found in dead wood in the examined provinces. I would rephrase it.

  • The sentence has been corrected.

Line 102: (PoznaÅ„, Poland) the period is missing at the end of the sentence “(PoznaÅ„, Poland).”

- The missing dot has been added.

3.Results

3.1. Geographical variation of Uropodina communities in Poland

Lines 105-107: I would rephrase it “52 Uropodina species were found in the examined merocenoses of the seven polish provinces. Only seven (13.5%) species occurred in all the provinces.”

  •  

Line 110: I would include in the text of the results also the percentages of the most abundant species across the provinces. It will give a clear picture of the findings at first glance.

  • The percentages of the most abundant species in the provinces have been

Line 112: Table 1 I would list the species in this table grouping them according to the Family. I would also recommend to change the page orientation from vertical to horizontal to make the table easier to read (the species name will be then on one line).

  • The species are arranged in order of decreasing percentage of occurrence in the examined provinces - at the top there are species present in all provinces, and at the bottom, those that are present only in one. Such an arrangement is more important than a systematic one, from the perspective of the problem discussed in the study because it shows the frequency of occurrence of particular species in the examined provinces. We leave the decision on the orientation of the Table to the editors of the journal.

Other suggestion: Figure 2 of the results already has ID abbreviations of the provinces. I would use the same here.

  • Abbreviations of the provinces has been added instead of full names.

Carefully read all the species and correct the scientific names and authorship!

It is better to use the accepted species name rather than the synonym. Some of the names are misspelled.

Please refer to the International Commission on Zoological Nomenclature.

After correcting this, please carefully check also the other results tables (Table 2 to Table 9)

Below some example:

  • Dinychus carinatus A.Berlese, 1903
  • Olodiscus minima (Kramer, 1882): Uropoda minima Kramer, 1882(accepted name),Olodiscus minimus Kramer, 1882(synonym)
  • Trachytes aegrota(Koch, 1841)
  • Trematurella elegans (Kramer, 1882): is this the correct attribution?
  • Dinychus woelkeiW.Hirschmann Zirngiebl-Nicol, 1969
  • Trachytes pauperior (Berlese, 1914)
  • Urodiaspis tecta(P.Kramer, 1876)
  • Dinychus arcuatus (Trägårdh, 1943)
  • Discourella (?) baloghi: why there is a question mark?
  • Polyaspis snasoneiBerlese, 1916. Polyaspis sansonei Berlese, 1916
  • …and so on…
  • Names of species and authors have been checked and corrected.

Lines 113-114: I would include for each following provinces paragraphs, the specification of the habitat were the species occurred.

  • The type of forest is not such important here for the species composition of Uropodina in the dead wood samples, as the material was not always collected from the dominant species in the tree stand, what is important is the species of tree it came from (Zacharyasiewicz et al. 2021, DOI: 10.3390/d13120609), and for this reason we have included an additional table with the list of trees from which the dead wood came in each province (Table A1).

Line 118: (Table 2 ). Remove space after 2

  • Space removed.

Lines 123, 135, 144, 153, 161, 170, 178Awer±SD. Correct the spelling. Be consistent in the table: if in the table caption are together, then in the table should be only one column and the SD should have the ± in front of the values.

  • Tables 2-8 and its captions have been corrected.

Line 135: Wielkopolskie or Wielkopolska? Be consistent all along the text.

N – number of specimens. In the table if the number is e.g. 9421 it should go all in the same line. Expand the size of the column.

  • Province names and Table 3 have been corrected.

Line 158: remove double space.

  • Corrected

Line 170: MaÅ‚opolskie or MaÅ‚opolska? Be consistent all along the text.

  • Corrected.

3.2. Geographical similarity of species composition of examined communities

Lines 188-191: communities inhabiting dead wood. Avoid to many repetitions

  • Corrected.

Lines 196-212: All the names are making the text hard to read and there is no correspondence with the codes showed by Figure 2. After naming the provinces the first time in the paragraph of the Study area, and introducing the ID codes, I suggest to use the same codes also in the text.

  • Acronyms of provinces name have been used in figures but in the text full names looks better in our opinion.

Line 214: Figure 1 I would use ID codes of the provinces to make the tree more compact and easier to read. Then, a legenda with the code names can be added in the figure description.

  • Figure has been changed.

Line 218: What kind of software was used to prepare these maps? Included it in the figure description

  • Information about software has been added in Materials and methods section. The map was generated using CorelDRAW 2020 (18) ((64Bit)—licence No. 382586, Poland, Poznan).

3.3. Species diversity of Uropodina communities in dead wood in some European countries

Lines 222-223: which collections? This should be mention in the Materials and methods paragraph.

  • All materials are in the same Collection at the Faculty of Biology, UAM, which is listed in Materials and methods.

Lines 225-226: Was the sampling method and the extraction across all the samples comparable? Is it possible that the low number of specimens in the wood samples is due to a different extraction procedure or a different preservation of the samples itself? Was the sampling method and the extraction similar to the polish samples?

  • All material was collected and shepherded using the same methods. The information used has been added in the text.

Line 228: the Netherlands

  • Corrected.

Line 229: only Norwegian samples are based on materials collected by the authors? This should be also specified in the Materials and methods.

  • All samples were collected by the authors. The information used has been added in the text.

Line 240: In Figure 3 A there is a yellow island (?) to the west of the Portugal coast. Please correct the figure.

  • The Figure 3 has been corrected.

I would enlarge the focus on the countries object of the study and remove all the meridians and parallels so that the picture will be more clear. The degrees on the frame are self-explanatory.

What kind of software was used to prepare these maps? Included it in the figure description

  • The maps show the different types of range of Uropodina species – the range of the common species, with wide distribution and those with a narrow range based on previous research. The map is from an earlier publication - the source is given in the caption.

Lines 243-244: Why unconfirmed occurrence is reported on the map with specimens found in the European dead wood merocenoses? I think this map is misplaced: it should not be at the end of the Results but as part of the Discussion. I would also make more evident the colour of the assessed geographical ranges in contrast with the unconfirmed occurrence (e.g. black/light grey, blu/yellow; red/yellow…).

  • Map F has been discarded.

4.Discussion

Line 245: insert space between 4. and Discussion

  •  

Line 257: aforementioned factors. Which factors? Nothing was previously mentioned.

  • We’ve discussed above about the influence of Pleistocene glaciations and the differences in the extent of the research in particular regions of Poland. Maybe the term "factors" is inappropriate here, it has been changed to "facts".

Lines 257-258: Without knowing the exact sampling procedure, it is difficult to assume anything. It is fundamental for the paper to be thorough and provide details of the data collection. Only then, it will be possible to discuss whether or not the modeling of the research is responsible for these findings.

  • The method of sample collection was the same in each case. The information used has been added in the text.

Lines 259-260: Quantity is not always the answer. A solid and standardized sampling system can also help in the data interpretation.

  • The collection methods were the same; what was not uniform was the degree of investigation and therefore the number of samples and the number of environments investigated. Only a large number of samples collected from different tree species can detect rare and scarce species.

Lines 262-263: not only due to their species composition, but also their community structure. Are you talking about the species composition and community structure of the Uropodina or of the unstable microhabitats?

  • The sentence has been rephrased.

Lines 264-265: was it ever consider that maybe this is connected to the food resources presents in different habitats? If so, I would cite some examples.

  • This distinct clustering structure may be related to nutritional resources, but also to different microhabitat conditions (i.e. moisture content, temperature) in the soil and microenvironments. This issue has not been studied already.

Lines 267-268: whole community [8]. A similar structure can be observed in the analysed  communities inhabiting dead wood from the seven surveyed provinces in Poland. The cited work (NapieraÅ‚a and BÅ‚oszyk, 2013) is also based on polish samples…so can you say that the structure is similar to itself? I would find another paper that covers other regions or I would think a better way to phrase these two sentences.

  • In the study by NapieraÅ‚a and BÅ‚oszyk (2013) different types of microhabitats are included, not only dead wood. The comparison of the results of the study from individual provinces with those from the whole of Poland and from different microhabitats, was intended to show that the same community structure is also present in individual regions. Unfortunately, there are no such studies on Uropodina from other regions of Poland or Europe.

Lines 269-272: the most dominant species were O. ovalis and P. pulchella but they were only dominant in 3 out of 7 provinces investigated? To better highlight the findings, I would rephrase the sentences not form provinces perspective but from the species perspective.

E.g. O. ovalis was one of the most dominant species in all seven provinces (from 19,15% up to 63,12%).

Reading the results, P. pulchella was one of the dominant species only in 3 provinces out of 7.

  • The sentence has been rephrased.

Line 274-278: more samples more species, less samples less species. How can you state that a low frequency had impact on the species diversity?? undoubtedly had an impact. What impact? In which way had an impact? Isn’t maybe a problem with the sampling?

This part of the discussion does not make sense.

  • The samples were collected in each of the provinces surveyed and the specimens were extracted using the same methods, so it is not the methodology that influences the low species diversity in some provinces, but the status of the survey and therefore the number of samples collected. In addition, this is also influenced by the low frequency of Uropodina in the samples.

Line 280-281, 283: species…species…species…species. Rephrase it.

  • The sentence has been rephrased.

Lines 283-284: of Uropodina communities inhabiting dead wood. I would remove it. This sentence is extremely frequent and makes the text tedious to read.

  • The sentence has been rephrased.

Line 284: other conditions found previously. What conditions? What are you referring to?

  • The sentence has been rephrased.

Lines 283-298: all this part should be in the Introduction! Nothing in the introduction was told about the ecology of the Uropodina! It is a fundamental part to interpretate the data correctly and be able to discuss it!

  • The study presented here is another, one of many on Uropodina communities inhabiting dead wood. In Introduction, there are numerous references to previous works on this problem (lines 43-55), so we felt that describing the ecology of Uropodina in dead wood, and therefore repeating the results of previous work, was not necessary here.

Lines 302-305: some species aren’t abbreviated. Why? Were all the species already mentioned at least one time in the previous text?

  • The full genus names of species which have not been mentioned before, have been added.

Lines 306-308: ?

  • We would be grateful for more clear comment.

Lines 311-316: some species aren’t abbreviated. Why? Were all the species already mentioned at least one time in the previous text?

  • The full genus name of species which has not been mentioned before, has been added.

Lines 319-339: through the entire paper no little relevance was given to European samples. If the results aren’t relevant and the samples not sufficient…why are included in the article. I would consider, at this point, to remove from the analysis the European samples all together or put a lot more effort in the description, the methods, the results and the discussion.

  • The purpose of including the samples from Europe was to show the status of the research of Uropodina in dead wood in the countries mentioned. The current state of the study does not allow a more detailed analysis, but the results obtained already indicate a similar trend as we observed in Poland, i.e. the dominance structure of the most common species is similar in all countries, while rare species, including endemics, appear only in some countries, and their detection requires more intensive research. We believe that this is an important conclusion because the role of faunal studies, is currently underestimated, while at the same time they are indispensable in monitoring environmental changes and protecting biodiversity.

  1. Conclusions

Based on the introduction and the discussion, the conclusions are also a bit weak.

Lines 344: The results presented here to show. Remove to.

  • The sentence has been rephrased.

Line 355: Author Contributions Please check carefully again. There are many mistakes in the punctuation.

  • Punctuation has been corrected.

Line 371: References Please check carefully the correspondence of the cited articles with the text and be consistent in the references format.

  • The list of references has been checked.

Round 2

Reviewer 2 Report

Authors has corrected some mistakes and added baseline data on their dataset essential for better evaluation and understanding of their work. Unfortunately, I am sorry to say that, but their revision was only superficial and did not address all major issues pointed by me and other Reviewers. Moreover, added new information suggests some further inconsistencies in Methodology.

For instance, from the map of sampling sites (Figure 1) it is clearly visible that a number of sampling sites was located in the Lubuskie region (western Poland). Data from Lubuskie are also discussed in the manuscript (see line 237) but none are presented in Results and manuscript lack any analyses of these data. Lubuskie is also not listed in the Method section. These inconsistencies have to be clarified. By the way, as this is clearly not that obvious for readers unfamiliar with geography and administrative division of Poland, adding borders of polish provinces to the map given in Figure 1 is a must. Moreover, Figure 1 contains a misty rectangle with some information which is completely unreadable (see the lower right corner) this has to be edited to make its content readable or removed if it is redundant.

I also have to point out again to usage of wrong terminology. As I have indicated previously using the term "community" for regional faunas is not appropriate, more it is wrong!. The term "community" always implies some kind of interactions among biota forming the community. In this case since authors have analyses mites from distant locations or even different geographical regions and countries, it is completely certain that there are no interactions among them at all. Thus the term "fauna" has to be used instead. Of course the manuscript contains some statements in which the term "community" can be used, but its usage for regional fauna or fauna of a particular country is unacceptable! Science it is not poetry but science is based on strict and rigorous methodology and clearly defined terminology!!! To my previous comment on that the authors has rebutted that such terminology is used in literature. Well, this is only partly true as such vague usage of that terminology is not found in any of the high quality international journals but only in some local low impact journals.

Taxonomists often point that ecologists do not distinguish or even often mistake investigated taxa. I do agree that this happens and it is unacceptable, but I would like to point out this is unacceptable in the same way as vague and incorrect usage of ecological terminology by taxonomists. Hope this comparison would make my point more clear and help the authors to understand that “community” is not a synonym for “fauna” of a region.

Regarding usage of terminology, one of the other reviewers pointed correctly that Poland is in Europe, so writing “in Poland and Europe” is incorrect. The authors have corrected that in the title and probably in some places in the text, but such expression is still found several times in the manuscript. For instance, just Abstract contains that phrase twice (see line: 15 & 22). Similarly, Conclusions also contain such statement twice (see line: 369 & 373). Other parts of the text have also be checked and corrected where necessary. Moreover statements such as “The total number of species … in the surveyed European countries was 24 species, ranging from 4 to 13 species per country” (e.g.; lines 20-21) are also false and it seems to place Poland on some other continent as 52 species were found there.

Authors have added data that clearly show very uneven sampling effort. Well, as they clearly pointed to that and discussed that this is acceptable. However, to demonstrate to what extend uneven sampling effort could have influenced the results analyses of number of samples and species richness would be necessary. Refraction curves (number of samples vs. number of species) would allow to estimate the total expected regional (country) diversity of the investigated mites and provide guidelines for the sampling effort optimization.

Finally, the whole text need truly detailed linguistic correction. Despite authors statement that the text was linguistically corrected, the English style and grammar is still of very poor quality. Well, I would say it is mostly understandable but it needs truly detailed linguistic correction.

Finally, I have also several other comments and/or suggestions that I have listed below. But I have to point out that the list, particularly regarding English style and grammar, is not exhausted and hope authors will put more serious effort into their manuscript correction.

Lines 31 & 59:

I would suggest replacing “zoogeography” by “biogeography”, both in the keywords and in the manuscript text.

Lines 49, 60 & more:

“mites in question” -> “investigated mites” reads better

Lines 68-69:

“in selected European countries, i.e. in the following countries: France, Italy, …”

The following reads better: “in selected (investigated) European countries, including France, Italy, …” but be aware of Poland which is also located in Europe.

Line 79:

“… (Figures 1, 3)”

In general, figures should be quoted in the text using consecutive numbers. As Figure 2 was not quoted in the text before that changing the figures numbering might be necessary.

Table 1.

I have previously pointed to inconsistency in the number of recorded species in Zachodniopomorskie given in Table 1 and in the text, Table 1 listed 17 species and text 16. Authors have corrected data in Table to 16. But in Zachodnipomorskie 17 species were recorded (!), so the text should be corrected not the data in Table 1. I am aware that one of the species was determined only to the genus level (i.e.; Pseudouropoda sp.), however as no other species from that particular genus was recorded in Zachodnipomorskie this is clearly an additional 17th species found in that region. Text the line 127 should be corrected accordingly.

Tables 2-8:

In English “Avg” is generally used as abbreviation for “average”.  

So, change “Aver” to “Avg”, in all Tables (both in Tables Captions and row 1).

Lines 214-216:

“The communities with the highest species found in Wielkopolskie are characterized by low similarity to the north-western areas of Zachodniopomorskie and Podkarpcakie, which is the most south-eastern province.”

This statement is both grammatically and logically incorrect. The first part of that statement implies that data from Wielkopolskie were compared with the north-western area of Zachodniopomorskie; that is only with the north-western part of Zachodniopomorskie. But as I guess authors intended to emphasize that Zachodniopomorskie is located in the north-western part of Poland. This whole sentence has to be rewritten.

Lines 237:

“Lubuskie” – where are data for the sampling sites in Lubuskie (shown in Figure 1)?

Line 254:

“the number of samples was always similar”

The following reads better: “the number of samples was relatively similar”

Lines 259-266:

If this is not a Table Footnote a space between the Table and that paragraph should be inserted.

Line 259:

“in the of dead wood”

Grammar correction needed.

Lines 259-260:

“in the European countries under scrutiny”

The following reads better: “in the investigated European countries”

Lines 265-266:

“The other 19 species occurred only once or twice in the analyzed material from the countries in question”

The following reads better: “The other 19 species occurred only once or twice”. The rest of that sentence is redundant. However, it has to be clearly explained what does “once or twice” refers. Does that refer to number of specimens (only 1-2 specimens found), samples (found only in 1-2 samples) or countries (or found only 1-2 countries).

Lines 282-285:

Grammar and style correction needed.

Line 302:

“what probably reducing”

Grammar correction needed.

Line 331:

“select provinces” -> “selected provinces”

Line 348:

“the European countries in question”

The following reads better: “the investigated European countries”

Lines 353-355:

Grammar and style correction needed.

Line 331:

“in her study”

Redundant addition.

I am truly looking forward to see that work to be published. But considering all the above I belive that this work still need revision. Hope all the comments and suggestions, both given by me and other reviewers, will help the authors improve the manuscript and they will consider them seriously.

 ------------------------------------------------------------------------------------

Author Response

Responses to Reviewer 2

Geographical differentiation of mites from the suborder Uropodina (Acari: Mesostigmata) in dead wood in Europe and Poland in the light of recent research

The authors of the study are grateful to the Reviewer for all comments and suggestions. All of them have been considered seriously and turned out to be extremely helpful, which obviously has considerably improved the overall quality of the manuscript. We hope that current version will be accepted by Reviewer.

Detailed responses to the Reviewer comments:

Authors has corrected some mistakes and added baseline data on their dataset essential for better evaluation and understanding of their work. Unfortunately, I am sorry to say that, but their revision was only superficial and did not address all major issues pointed by me and other Reviewers. Moreover, added new information suggests some further inconsistencies in Methodology.

For instance, from the map of sampling sites (Figure 1) it is clearly visible that a number of sampling sites was located in the Lubuskie region (western Poland). Data from Lubuskie are also discussed in the manuscript (see line 237) but none are presented in Results and manuscript lack any analyses of these data. Lubuskie is also not listed in the Method section. These inconsistencies have to be clarified. By the way, as this is clearly not that obvious for readers unfamiliar with geography and administrative division of Poland, adding borders of polish provinces to the map given in Figure 1 is a must. Moreover, Figure 1 contains a misty rectangle with some information which is completely unreadable (see the lower right corner) this has to be edited to make its content readable or removed if it is redundant.

  • The map has been corrected - the borders of the provinces have been marked and points from Lubuskie, which were marked previously by mistake, have been removed. Also unnecessary mention of this province in Results was omitted, as it is not the subject of the research.

I also have to point out again to usage of wrong terminology. As I have indicated previously using the term "community" for regional faunas is not appropriate, more it is wrong!. The term "community" always implies some kind of interactions among biota forming the community. In this case since authors have analyses mites from distant locations or even different geographical regions and countries, it is completely certain that there are no interactions among them at all. Thus the term "fauna" has to be used instead. Of course the manuscript contains some statements in which the term "community" can be used, but its usage for regional fauna or fauna of a particular country is unacceptable! Science it is not poetry but science is based on strict and rigorous methodology and clearly defined terminology!!! To my previous comment on that the authors has rebutted that such terminology is used in literature. Well, this is only partly true as such vague usage of that terminology is not found in any of the high quality international journals but only in some local low impact journals.

Taxonomists often point that ecologists do not distinguish or even often mistake investigated taxa. I do agree that this happens and it is unacceptable, but I would like to point out this is unacceptable in the same way as vague and incorrect usage of ecological terminology by taxonomists. Hope this comparison would make my point more clear and help the authors to understand that “community” is not a synonym for “fauna” of a region.

  • The term „community” has been replaced with „fauna”.

Regarding usage of terminology, one of the other reviewers pointed correctly that Poland is in Europe, so writing “in Poland and Europe” is incorrect. The authors have corrected that in the title and probably in some places in the text, but such expression is still found several times in the manuscript. For instance, just Abstract contains that phrase twice (see line: 15 & 22). Similarly, Conclusions also contain such statement twice (see line: 369 & 373). Other parts of the text have also be checked and corrected where necessary. Moreover statements such as “The total number of species … in the surveyed European countries was 24 species, ranging from 4 to 13 species per country” (e.g.; lines 20-21) are also false and it seems to place Poland on some other continent as 52 species were found there.

  • The expressions indicated by the reviewer are due to the fact that Poland was discussed separately in the paper, as the state of research in this country is more advanced and the material could have been analysed more thoroughly. However, they could be misleading indeed and have therefore been corrected where it was necessary.

Authors have added data that clearly show very uneven sampling effort. Well, as they clearly pointed to that and discussed that this is acceptable. However, to demonstrate to what extend uneven sampling effort could have influenced the results analyses of number of samples and species richness would be necessary. Refraction curves (number of samples vs. number of species) would allow to estimate the total expected regional (country) diversity of the investigated mites and provide guidelines for the sampling effort optimization.

  • Drawing Refraction curves is impossible and would be factually incorrect because the samples come from different study periods, different provincial regions, with very different species diversity and different types of dead wood (logs, stumps, hollows). For example, in Podlaskie, some samples were collected in the BiaÅ‚owieża Forest, where species diversity is very high and it may happen that a few samples from such a rich area will enable to collect the whole or most of the species pool from the province. The same applies to the Cisy Staropolskie im. L. WyczóÅ‚kowski Reserve in Wierzchlas (in Kujawsko-Pomorskie), which is also an exceptionally faunistically diverse and rich area. On the example of the BiaÅ‚owieża Primeval Forest, it was also shown that collecting samples from new areas of this complex results in recording of new species (data not yet published). Therefore, it is not possible to indicate the minimum/optimal number of samples for each province that should be collected. This would be possible if we carried out the study in a homogeneous environment, but in reality the diversity of areas, both in terms of microenvironments and other characteristics, is too high to do so.

Finally, the whole text need truly detailed linguistic correction. Despite authors statement that the text was linguistically corrected, the English style and grammar is still of very poor quality. Well, I would say it is mostly understandable but it needs truly detailed linguistic correction.

Finally, I have also several other comments and/or suggestions that I have listed below. But I have to point out that the list, particularly regarding English style and grammar, is not exhausted and hope authors will put more serious effort into their manuscript correction.

Lines 31 & 59:

I would suggest replacing “zoogeography” by “biogeography”, both in the keywords and in the manuscript text.

Lines 49, 60 & more:

“mites in question” -> “investigated mites” reads better

Lines 68-69:

“in selected European countries, i.e. in the following countries: France, Italy, …”

The following reads better: “in selected (investigated) European countries, including France, Italy, …” but be aware of Poland which is also located in Europe.

Line 79:

“… (Figures 1, 3)”

In general, figures should be quoted in the text using consecutive numbers. As Figure 2 was not quoted in the text before that changing the figures numbering might be necessary.

- All the above mentioned corrections suggested by the reviewer have been made and the text has been proofread and linguistically corrected.

Table 1.

I have previously pointed to inconsistency in the number of recorded species in Zachodniopomorskie given in Table 1 and in the text, Table 1 listed 17 species and text 16. Authors have corrected data in Table to 16. But in Zachodnipomorskie 17 species were recorded (!), so the text should be corrected not the data in Table 1. I am aware that one of the species was determined only to the genus level (i.e.; Pseudouropoda sp.), however as no other species from that particular genus was recorded in Zachodnipomorskie this is clearly an additional 17th species found in that region. Text the line 127 should be corrected accordingly.

  • In Zachodniopomorskie there are 16 species (Table 2). In Table 1, Urodiaspis pannonica, which did not occur there, was marked by mistake.

Tables 2-8:

In English “Avg” is generally used as abbreviation for “average”.  

So, change “Aver” to “Avg”, in all Tables (both in Tables Captions and row 1).

  •  

Lines 214-216:

“The communities with the highest species found in Wielkopolskie are characterized by low similarity to the north-western areas of Zachodniopomorskie and Podkarpcakie, which is the most south-eastern province.”

This statement is both grammatically and logically incorrect. The first part of that statement implies that data from Wielkopolskie were compared with the north-western area of Zachodniopomorskie; that is only with the north-western part of Zachodniopomorskie. But as I guess authors intended to emphasize that Zachodniopomorskie is located in the north-western part of Poland. This whole sentence has to be rewritten.

  • The sentence has been corrected.

Lines 237:

“Lubuskie” – where are data for the sampling sites in Lubuskie (shown in Figure 1)?

- The data from Lubuskie were not included in the study. Locations from this province were marked on the map by mistake (they have been removed in the new version).

Line 254:

“the number of samples was always similar”

The following reads better: “the number of samples was relatively similar”

  •  

Lines 259-266:

If this is not a Table Footnote a space between the Table and that paragraph should be inserted.

  • The space has been added.

Line 259:

“in the of dead wood”

Grammar correction needed.

  •  

Lines 259-260:

“in the European countries under scrutiny”

The following reads better: “in the investigated European countries”

  • Corrected.

Lines 265-266:

“The other 19 species occurred only once or twice in the analyzed material from the countries in question”

The following reads better: “The other 19 species occurred only once or twice”. The rest of that sentence is redundant. However, it has to be clearly explained what does “once or twice” refers. Does that refer to number of specimens (only 1-2 specimens found), samples (found only in 1-2 samples) or countries (or found only 1-2 countries).

  • The sentence has been corrected. These 19 species have been found “once or twice” in all material from these countries.

Lines 282-285:

Grammar and style correction needed.

  •  

Line 302:

“what probably reducing”

Grammar correction needed.

  • Corrected.

Line 331:

“select provinces” -> “selected provinces”

  • Corrected.

Line 348:

“the European countries in question”

The following reads better: “the investigated European countries”

  • Corrected.

Lines 353-355:

Grammar and style correction needed.

  • Corrected.

Line 331:

“in her study”

Redundant addition.

  • Corrected

I am truly looking forward to see that work to be published. But considering all the above I belive that this work still need revision. Hope all the comments and suggestions, both given by me and other reviewers, will help the authors improve the manuscript and they will consider them seriously.

 ------------------------------------------------------------------------------------

Reviewer 3 Report

I appreciate many of the changes and I feel the manuscript is quite improved.

However, I think that the Introduction still has to be improved by expanding the ecological importance of the dead wood habitats and the Uropodina communities. Moreover, a careful selection of the literature has to be done in the Introduction paragraph. The substantial amount of self-citation can be easily reduced for example by removing the publications not widely accessible to the scientific community (e.g. reference 5 does not even show any correspondence on the freely accessible web search engines such as Google Scholar; the same for reference 16) and selecting only a few relevant articles on the topic, possibly in English.

Going through the authors' rebuttal and the manuscript, I had the feeling that little effort has been made to provide a neater version of the work. Frustratingly, I found out that some of my previous recommendations have been addressed superficially. I am especially referring to the wide use of synonyms instead of the accepted scientific names for some species. These synonyms have been used for years by the authors, also in other publications, although the accepted nomenclature is available, giving the impression of the authors attempting to increase their citation rate artificially.

I strongly advise refraining from such practices and to use the correct names according to the accepted nomenclature.

I also recommend carefully reading the draft and correcting all the consistency issues when come to the use of spaces and punctuation before the submission. With a neat draft, it is easier to focus on the manuscript's content without distractions.

14: was to assess change in “is to assess”

18: from the Polish samples

36-40: I would avoid using names. The citation of the work combined with the reference is already enough.

56: missing citation

57: decaying wood

58: remove double space

66: Probably better to use a word like decaying while writing about wood (check along the text).

73-79: I suggest rephrasing it as follows: The material for this study comprises 1369 samples of dead wood (lying trunks, stumps, and hollows), collected in Poland (1180) and France (25), Italy (20), Belgium (23), the Netherlands (20), Slovakia (27), Sweden (23), Norway (25), and Denmark (26) (Appendix 1). The material collected in Poland comes from seven geographically different regions of the country (i.e. Zachodniopomorskie (28), Wielkopolskie (694), Kujawsko-Pomorskie (81), Podlaskie (147), DolnoÅ›lÄ…skie (122), Podkarpackie (53), and MaÅ‚opolskie (55)) (Figures 1, 3).

81: Figure 1: remove the square next to the map.

87: and co-workers

90-94: the identification was conducted before the clearing with lactic acid?

99: remove double “:”

107: remove double “.”

Table 1: In my previous review, I asked the authors to carefully check all the scientific names of the species but little effort has been made. The scientific names used in the list have to be accepted names and not synonyms. A synonym is the outdated version of the current valid taxon name. I suggest making use also of the Global Biodiversity Information Facility ID (GBIF https://www.gbif.org) to check the nomenclature in order to avoid bringing on a tradition of erroneous names and avoiding self-citing their own works as a means of establishing a “new” standard.

e.g. Leiodinychus orbicularis: the accepted name is Trichouropoda orbicularis (Koch, 1839); Olodiscus minima(Kramer, 1882): the accepted name is Uropoda minima Kramer, 1882; Polyaspinus cylindricus Berlese, 1916: the accepted name is Uroseius cylindricus (Berlese, 1916)

202: II – moderate (od 50 do 60%). I suppose this is Polish for from…to. Please correct

288: remove the space before the comma

294: remove double “.”

295: remove double “.”

302: remove double “,”

312: remove double space

314: remove double “.”

338: remove double space

340: remove the space before the comma

351: remove the space before the comma

358-359: “in the dead wood of beech forests in France, as described by Athias-Binche [1,2] in her study…”. I would avoid using names. The citation is already enough.

389: Data Availability Statement: The link provided does not lead to the data presented in this study but to a general Polish website. Please, provide the right link to the data.

Appendix

Appendix 1

The way these data are presented has a lot of consistency issues when come to the use of spaces and punctuation (e.g. 1.Poppel3369E).Dead wood; diverserotten; Leg. J. BÅ‚oszyk.; Leg J. BÅ‚oszyk).

The data would be better organised in a table with the following columns:

Country; Sample ID; Data of collection; Coordinates; Area; Description; Notes

Be consistent when writing coordinates (or decimals or degrees) and pay attention to the format.

e.g. 51o38’35”N 05o03’30’E; 50.8157N 4.3318E

Check again the spelling of all the locations. E.g. San Vincento: correct spelling is San Vincenzo; Prowincja Siena: should be Siena provinceVicinty of Lund: if you provide coordinates it is not necessary to say vicinity.

Table A1: I would add a new column with the common names of the tree species or I suggest removing them altogether.

References: see comments above and please check carefully the text, some mistakes in the titles and names of the authors are present.

Author Response

Responses to Reviewer 3

Geographical differentiation of mites from the suborder Uropodina (Acari: Mesostigmata) in dead wood in Europe and Poland in the light of recent research

The authors of the study are grateful to the Reviewer for all comments and suggestions. All of them have turned out to be extremely helpful, which obviously has considerably improved the overall quality of the manuscript. We hope that current version will be accepted by Reviewer.

Detailed responses to the Reviewer comments:

I appreciate many of the changes and I feel the manuscript is quite improved.

However, I think that the Introduction still has to be improved by expanding the ecological importance of the dead wood habitats and the Uropodina communities. Moreover, a careful selection of the literature has to be done in the Introduction paragraph. The substantial amount of self-citation can be easily reduced for example by removing the publications not widely accessible to the scientific community (e.g. reference 5 does not even show any correspondence on the freely accessible web search engines such as Google Scholar; the same for reference 16) and selecting only a few relevant articles on the topic, possibly in English.

  • Introduction has been completed and the relevant citations have been added. However, we believe that some of the works cited are important to the topic discussed in this section, and therefore they cannot be omitted simply because they are more difficult to access or because they are not in English. Moreover, even if the publication in not accessible in the popular databases we may contact with the author and ask for the copy.

Going through the authors' rebuttal and the manuscript, I had the feeling that little effort has been made to provide a neater version of the work. Frustratingly, I found out that some of my previous recommendations have been addressed superficially. I am especially referring to the wide use of synonyms instead of the accepted scientific names for some species. These synonyms have been used for years by the authors, also in other publications, although the accepted nomenclature is available, giving the impression of the authors attempting to increase their citation rate artificially.

I strongly advise refraining from such practices and to use the correct names according to the accepted nomenclature.

I also recommend carefully reading the draft and correcting all the consistency issues when come to the use of spaces and punctuation before the submission. With a neat draft, it is easier to focus on the manuscript's content without distractions.

14: was to assess change in “is to assess”

  • Corrected.

18: from the Polish samples

  • We think that previous version sounds better.

36-40: I would avoid using names. The citation of the work combined with the reference is already enough.

  • We believe that in this case it is important to mention the name, since we are talking about the researcher who pioneered this line of research on Uropodina.

56: missing citation

  • In the sentence: “These observations, as well as the results of other studies on the zoogeography biogeography of this group of mites, have shown that many species of the investigated mites in question have different range of occurrence in Poland [5,9,26].” , the citations are given now at the end.

57: decaying wood

  • Corrected.

58: remove double space

66: Probably better to use a word like decaying while writing about wood (check along the text).

  • The term 'dead wood' is commonly used in the acarological literature, so we do not wish to remove it throughout the text, but it has been replaced with the term 'decaying' to avoid too many repetitions.

73-79: I suggest rephrasing it as follows: The material for this study comprises 1369 samples of dead wood (lying trunks, stumps, and hollows), collected in Poland (1180) and France (25), Italy (20), Belgium (23), the Netherlands (20), Slovakia (27), Sweden (23), Norway (25), and Denmark (26) (Appendix 1). The material collected in Poland comes from seven geographically different regions of the country (i.e. Zachodniopomorskie (28), Wielkopolskie (694), Kujawsko-Pomorskie (81), Podlaskie (147), DolnoÅ›lÄ…skie (122), Podkarpackie (53), and MaÅ‚opolskie (55)) (Figures 1, 3).

  • The paragraph has been replaced.

81: Figure 1: remove the square next to the map.

  • The square has been removed, and the map changed.

87: and co-workers

  • Corrected.

90-94: the identification was conducted before the clearing with lactic acid?

  • No, temporary slides was done only in case of some (mainly rare species) and some juvenile stages. Most of species was identified with a stereoscopic microscope.

99: remove double “:”

  • Corrected.

107: remove double “.”

  • Corrected.

Table 1: In my previous review, I asked the authors to carefully check all the scientific names of the species but little effort has been made. The scientific names used in the list have to be accepted names and not synonyms. A synonym is the outdated version of the current valid taxon name. I suggest making use also of the Global Biodiversity Information Facility ID (GBIF https://www.gbif.org) to check the nomenclature in order to avoid bringing on a tradition of erroneous names and avoiding self-citing their own works as a means of establishing a “new” standard.

e.g. Leiodinychus orbicularis: the accepted name is Trichouropoda orbicularis (Koch, 1839); Olodiscus minima(Kramer, 1882): the accepted name is Uropoda minima Kramer, 1882; Polyaspinus cylindricus Berlese, 1916: the accepted name is Uroseius cylindricus (Berlese, 1916)

 - In the case of this group of mites, two independent systematics have been in use since the 1960s. The nomenclature suggested by the Reviewer is the system developed by Hirschman and his co-workers. His Gangsystematic has been criticised from the outset by major acarological authorities as being incompatible with the Code of Zoological Nomenclature. The authors completely agree with the statement: "The systematics of the Uropodina then entered a dramatic new phase, with the arrival of Werner Hirschmann and his school, beginning in 1957." (Halliday R.B. 2016: "Catalogue of families and their type genera in the mite suborder Uropodina (Acari: Mesostigmata)". Zootaxa 4061 (4): 347-366)). At the same time, we would like to point out that the nomenclature used by the authors has been in place in the acarological literature for years and is consistent with the "Catalogue of genera and their type species in the mite suborder Uropodina (Acari: Mesostigmata" (Halliday 2015), and will therefore not be changed for the purposes of this publication.

202: II – moderate (od 50 do 60%). I suppose this is Polish for from…to. Please correct

  • Corrected.

288: remove the space before the comma

294: remove double “.”

295: remove double “.”

302: remove double “,”

312: remove double space

314: remove double “.”

338: remove double space

340: remove the space before the comma

351: remove the space before the comma

  • Spaces and punctuation have been corrected, where it was possible. However, the final editorial corrections will be made on the proof copy.

358-359: “in the dead wood of beech forests in France, as described by Athias-Binche [1,2] in her study…”. I would avoid using names. The citation is already enough.

  • Corrected.

389: Data Availability Statement: The link provided does not lead to the data presented in this study but to a general Polish website. Please, provide the right link to the data.

- The link has been replaced with one redirecting to the trial catalogue with the list of samples.

Appendix

Appendix 1

The way these data are presented has a lot of consistency issues when come to the use of spaces and punctuation (e.g. 1.Poppel3369E).Dead wood; diverserotten; Leg. J. BÅ‚oszyk.; Leg J. BÅ‚oszyk).

The data would be better organised in a table with the following columns:

Country; Sample ID; Data of collection; Coordinates; Area; Description; Notes

Be consistent when writing coordinates (or decimals or degrees) and pay attention to the format.

e.g. 51o38’35”N 05o03’30’E; 50.8157N 4.3318E

Check again the spelling of all the locations. E.g. San Vincento: correct spelling is San Vincenzo; Prowincja Siena: should be Siena provinceVicinty of Lund: if you provide coordinates it is not necessary to say vicinity.

  • The list of samples in the Appendix has been checked and corrected.

Table A1: I would add a new column with the common names of the tree species or I suggest removing them altogether.

  • We have decided to leave the sample list in its current form, as this is a commonly used and legible format for the sample list.

References: see comments above and please check carefully the text, some mistakes in the titles and names of the authors are present.

  • References section has been checked and corrected.

Round 3

Reviewer 2 Report

While authors significantly improved their manuscript I do have impression that this correction was kind of hastily and superficial. For instance, they seem to limit their correction to inconsistencies and mistakes that were pointed out, but left identical mistakes in other parts of the text. See data given in Line 286 (Discussion) where it is written that Uropodina fauna in Zachodniopomorskie consisted 17 species. This should be correct to 16 as it is given in Table 1 and in Results (Line 135). Consistency of other data between the text and Tables should also be check carefully.

However, the biggest problem is the English style as in some parts of the text it is still of a poor quality. Well, the text is understandable, but the manuscript would surely benefit greatly from professional English editing. The text surely needs also some editorial editing, for instance it contains a number of redundant spaces, or sometimes opposite it is lacking spaces. Such editing is, however, beyond reviewers role, so I will leave that to authors and editors. After these corrections I believe the manuscript should be ready for publication.

Author Response

Responses to Reviewer 2

Geographical differentiation of mites from the suborder Uropodina (Acari: Mesostigmata) in dead wood in Europe and Poland in the light of recent research

The authors of the study are grateful to the Reviewer for all comments and suggestions. All of them have been considered seriously and turned out to be extremely helpful, which obviously has considerably improved the overall quality of the manuscript. We hope that current version will be accepted by Reviewer.

Detailed responses to the Reviewer comments:

While authors significantly improved their manuscript I do have impression that this correction was kind of hastily and superficial. For instance, they seem to limit their correction to inconsistencies and mistakes that were pointed out, but left identical mistakes in other parts of the text. See data given in Line 286 (Discussion) where it is written that Uropodina fauna in Zachodniopomorskie consisted 17 species. This should be correct to 16 as it is given in Table 1 and in Results (Line 135). Consistency of other data between the text and Tables should also be check carefully.

Mistake in the Discussion has been corrected, and consistency of data between the text and tables has been checked.

However, the biggest problem is the English style as in some parts of the text it is still of a poor quality. Well, the text is understandable, but the manuscript would surely benefit greatly from professional English editing. The text surely needs also some editorial editing, for instance it contains a number of redundant spaces, or sometimes opposite it is lacking spaces. Such editing is, however, beyond reviewers role, so I will leave that to authors and editors. After these corrections I believe the manuscript should be ready for publication.

The article was double-checked and proofread by someone who has a PhD in English linguistics and has been translating and doing proofreading of our articles and research papers for over 10 years. Some of these studies have been published in other journals published by MDPI, i.e. Diversity or Forests. In addition, the work has been proofread by a native speaker. It is then difficult to expect more professional proofreading, and the stylistic side always remains a matter of taste.

As far as editorial errors are concerned (excess or missing spaces, etc.), the paper is in its current form in draft form with numerous corrections in Track Changes, so it is difficult to see all editorial errors. We will do our best to remove all these errors when the article has been proofread.

Reviewer 3 Report

The manuscript has greatly improved from its first form and I recognise the authors' efforts.

Unfortunately, I do not believe that the paper is publishable in its current form. There are two main reasons for my decision.

First, I agree with the authors that the systematics of the Uropodina is complex and this is not the appropriate article and Journal to tackle the problem. But in a scientific peer-reviewed manuscript one should always use the right and accepted name. The exception, the use of the synonym, can be allowed only with good argumentation, for instance with a thorough re-examination of the type specimens or the election of neotypes, preferably from the type locality.

In my previous reply, I underline the fact (as an example) that the name Leiodinychus orbicularis (C.L. Koch, 1839) is not the accepted name for, what I presume is, Trichouropoda orbicularis (Koch, 1839). The name Leiodinychus orbicularis does not even pop up as a combination for the genus Leiodinychus, and one of the authors of the current manuscript is the only one in the scientific literature using this odd combination.

This lead to the second reason, which I also pointed out in my previous reply, namely the practice of establishing a new name for a species by self-citing their previous works.

I believe that this manuscript shows manipulation of the citations and therefore is not aligned with the Citation Policy of the Journal and with the COPE guidelines.

Unfortunately, in its current state, I cannot recommend accepting this paper.

Author Response

Responses to Reviewer 3

Geographical differentiation of mites from the suborder Uropodina (Acari: Mesostigmata) in dead wood in Europe and Poland in the light of recent research

The authors of the study are grateful to the Reviewer for all comments and suggestions. All of them have turned out to be extremely helpful, which obviously has considerably improved the overall quality of the manuscript. We hope that current version will be accepted by Reviewer.

Detailed responses to the Reviewer comments:

The manuscript has greatly improved from its first form and I recognise the authors' efforts.

Unfortunately, I do not believe that the paper is publishable in its current form. There are two main reasons for my decision.

First, I agree with the authors that the systematics of the Uropodina is complex and this is not the appropriate article and Journal to tackle the problem. But in a scientific peer-reviewed manuscript one should always use the right and accepted name. The exception, the use of the synonym, can be allowed only with good argumentation, for instance with a thorough re-examination of the type specimens or the election of neotypes, preferably from the type locality.

In my previous reply, I underline the fact (as an example) that the name Leiodinychus orbicularis (C.L. Koch, 1839) is not the accepted name for, what I presume is, Trichouropoda orbicularis (Koch, 1839). The name Leiodinychus orbicularis does not even pop up as a combination for the genus Leiodinychus, and one of the authors of the current manuscript is the only one in the scientific literature using this odd combination.

As far as the issue of the systematic is concerned, the authors are aware of the systematic problems within the suborder Uropodina, and have consistently refrained from using the systematics of the Hirschmann school since the beginning of their research, recognising the validity of its criticism by major acarologists. As our work from Poland was never taxonomic in nature, but concerned the biology, ecology and zoogeography of known species and referred mainly to the communities of these mites, we used the names commonly accepted by Evans and Athias-Binche, sometimes only correcting the generic affiliation in the species recognised by Hirschmann and his co-workers or occasionally only synomising them (with obvious grounds for doing so).

In 1979, in the 26th volume of Acarologie, a journal he published (nota bene not reviewed by independent reviewers), Hirschmann presented the concept of a new systematics similar to that used by other acarologists. It seemed that at this point the taxonomic problems within this group of mites would end. Unfortunately, this 'Stadiensystem' was abandoned by Hirschmann and his colleagues and there was a return to the criticised 'Gangsystem'. It is in the publication Stadien Systematikder Parsitiformes Teil I on page 64 that the genus Leiodinychus BERLESE 1917 appears instead of Orbicularis-Gruppe with the type species Notaspis orbicularis C.L. Koch 1839. Koch's species was recognised by Hirschmann and Zirgiebl-Nicol in 1961 and included in the genus Trichouropoda. Meanwhile, the genus Trichouropoda should only be used to refer to species such as Trichouropoda bipilis and related species, characterised by a pair of very long setae below the anal opening (1916 - Uropoda (Trichouropoda) longiseta Berlese, Redia 12:142). In 1882, G. Canestrini describes the species Uropoda krameri. (Acarofauna Italiana 105) considered by Hirschamnna to be a junior synonym of the species described by Koch. In 1917 Berlese used for the first time the name Leiodinychus as a subgenus for this species (1917 Urodinychus (Leiodinychus) krameri Berlese, Redia 13:12), and since 1934 the name 1934 - Leiodinychus krameri has regularly appeared in the literature (André, Bull. Soc. Zool. 59(2):112-122 (Figs. 1-6)) BÅ‚oszyk and Athias-Binche considered that in classifying the aforementioned species in the Berlesian genus Leiodinychus, the species name given earliest by Koch, i.e. 'orbicularis' should be retained rather than 'krameri'. This is where this name originated and is consistently used by us to this day.

As the systematics of Uropodina needs to be completely revised, we hope that someone will one day undertake to 'clean up this Augean  Stables'. At present, we have been using the same nomenclature for more than 40 years, mainly so that future researchers, when carrying out a systematic revision, can easily use all the information on the biology, ecology, distribution of a species and its importance in the Uropodina communities without wondering which species is meant and whether it has been identified correctly. In addition, all the material we have studied so far is deposited in the museum facility "Natural History Collections" at the Faculty of Biology of the Adam Mickiewicz University in PoznaÅ„ and can be made available at any time to any interested researcher to verify the designations.

This lead to the second reason, which I also pointed out in my previous reply, namely the practice of establishing a new name for a species by self-citing their previous works.

I believe that this manuscript shows manipulation of the citations and therefore is not aligned with the Citation Policy of the Journal and with the COPE guidelines.

The paper is based on previous research, so it is difficult to dissociate ourselves from it and not cite our previous work that relates to this topic, especially as this issue has not been addressed by other authors. We have removed from the references two publications that are not strictly related to the topic under discussion, but citing others that deal with Uropodina in dead wood or the geographical distribution of these mites is considered necessary.
